# MAX-DOAS measurements of tropospheric NO$_2$ and HCHO in Nanjing and the comparison to OMI observations

Ka Lok Chan[1], Zhuoru Wang[1], Aijun Ding[2], Klaus-Peter Heue[1], Yicheng Shen[2], Jing Wang[3], Feng Zhang[4], Yining Shi[4], Nan Hao[5], and Mark Wenig[6]

[1]Remote Sensing Technology Institute (IMF), German Aerospace Center (DLR), Oberpfaffenhofen, Germany
[2]School of Atmospheric Sciences, Nanjing University, Nanjing, China
[3]Key Laboratory for Aerosol-Cloud-Precipitation of China Meteorological Administration, Nanjing University of Information Science and Technology, Nanjing, China
[4]Key Laboratory of Meteorological Disaster, Ministry of Education, Nanjing University of Information Science and Technology, Nanjing, China
[5]European Organisation for the Exploitation of Meteorological Satellites (EUMETSAT), Darmstadt, Germany
[6]Meteorological Institute, Ludwig-Maximilians-Universität München, München, Germany

**Correspondence:** Ka Lok Chan (ka.chan@dlr.de)

**Abstract.**

In this paper, we present long term observations of atmospheric nitrogen dioxide (NO$_2$) and formaldehyde (HCHO) in Nanjing using a Multi-AXis Differential Optical Absorption Spectroscopy (MAX-DOAS) instrument. Ground based MAX-DOAS measurements were performed from April 2013 to February 2017. The MAX-DOAS measurements of NO$_2$ and HCHO

vertical column densities (VCDs) are used to validate OMI satellite observations over Nanjing. The comparison shows that the OMI observations of NO$_2$ correlate well with the MAX-DOAS data with Pearson correlation coefficient ($R$) of 0.91. However, OMI observations are on average a factor of 3 lower than the MAX-DOAS measurements. Replacing the a priori NO$_2$ profiles by the MAX-DOAS profiles in the OMI NO$_2$ VCD retrieval would increase the OMI NO$_2$ VCDs by $\sim$30 % with correlation nearly unchanged. The comparison result of MAX-DOAS and OMI observations of HCHO VCD shows a

good agreement with $R$ of 0.75 and the slope of the regression line is 0.99. An age weighted backward propagation approach is applied to the MAX-DOAS measurements of NO$_2$ and HCHO to reconstruct the spatial distribution of NO$_2$ and HCHO over Yangtze River Delta during summer and winter time. The reconstructed NO$_2$ fields show a distinct agreement with OMI satellite observations. However, due to the short atmospheric lifetime of HCHO, the backward propagated HCHO data does not show a strong spatial correlation with the OMI HCHO observations. The result shows the MAX-DOAS measurements are

sensitive to the air pollution transportation in the Yangtze River Delta, indicating the air quality in Nanjing is significantly influenced by regional transportation of air pollutants. The MAX-DOAS data are also used to evaluate the effectiveness of air pollution control measures implemented during the Youth Olympic Games 2014. The MAX-DOAS data show a significant reduction of ambient aerosol, NO$_2$ and HCHO (30 % - 50 %) during the Youth Olympic Games. Our results provide a better understanding of the transportation and sources of pollutants in over the Yangtze River Delta as well as the effect of emission

control measures during large international event, which are important for the future design of air pollution control policies.

# 1 Introduction

Nitrogen dioxide ($NO_2$) and formaldehyde (HCHO) are major atmospheric pollutants playing crucial roles in atmospheric chemistry. $NO_2$ is a catalyst for ozone ($O_3$) formation in the troposphere, while also participating in the catalytic destruction of stratospheric $O_3$ (Crutzen, 1970). Major $NO_2$ sources include fossil fuel combustion, biomass burning, lightning and oxidation of ammonia (Bond et al., 2001; Zhang et al., 2003). The emission of $NO_x$ shows a significant increasing trend in China due to the rapid industrialization and economy development in the last two decades (Zhang et al., 2007; van der A et al., 2008; Zhao et al., 2013), making it one of the most severe air pollution problems. HCHO is an intermediate product of the oxidation of almost all volatile organic compounds (VOCs). Therefore, it is widely used as an indicator of non methane volatile organic compounds (NMVOCs) (Fried et al., 2011). VOCs also have significant impacts on the abundance of hydroxyl (OH) radicals in the atmosphere, which is the major oxidant in the tropospheric. Major HCHO sources over the continents include the oxidation of VOCs emitted from plants, biomass burning, traffic and industrial emissions. Oxidation of methane ($CH_4$) emitted from the ocean is the main source of HCHO over water. Both $NO_2$ and HCHO contribute to the formation of secondary aerosols (Jang and Kamens, 2001), while the contribution from HCHO is relatively small. $NO_2$ and HCHO are also toxic to human in high concentration. The spatial distribution of $NO_2$ and HCHO is strongly related to their emissions due to their short atmospheric lifetime. Consequently, it is important to understand the spatial and temporal variations of atmospheric $NO_2$ and HCHO for better air pollution management and control.

Multi-AXis Differential Optical Absorption Spectroscopy (MAX-DOAS) is a powerful remote sensing measurement technique which provides valuable vertical distribution information of atmospheric aerosols and trace gases (Platt and Stutz, 2008). Information of tropospheric aerosols and traces gases are obtained from the molecular absorption in the ultraviolet and visible spectral bands by applying the differential optical absorption spectroscopy (DOAS) technique to the observations of scattered sun light spectrum in several different viewing directions. As the experimental setup of MAX-DOAS is rather simple and inexpensive, it has been widely used for the observation of atmospheric aerosols and trace gases in the past decade (Hönninger and Platt, 2002; Hönninger et al., 2004; Wittrock et al., 2004; Frieß et al., 2006; Irie et al., 2008; Li et al., 2010; Clémer et al., 2010; Halla et al., 2011; Li et al., 2013; Ma et al., 2013; Chan et al., 2015; Jin et al., 2016; Wang et al., 2016; Chan et al., 2018). MAX-DOAS measurements are highly sensitive to aerosols and trace gases in the lower troposphere and provide valuable information of the vertical distribution of aerosol extinction and trace gases. The information of the aerosol and trace gas vertical profiles are particularly important for the study of the physical and chemical processes in the atmosphere.

Satellite based remote sensing measurements provide indispensable spatial information of air pollutants (Burrows et al., 1999; Bovensmann et al., 1999; Callies et al., 2000; Levelt et al., 2006). Trace gas columns are derived from the satellite observations of Earth's reflected solar spectrum for the investigation of atmospheric dynamics and emissions from both anthropogenic and natural sources (Beirle et al., 2003; Wenig et al., 2003; Beirle et al., 2004; Richter et al., 2005; Zhang et al., 2007; van der A et al., 2008). Satellite measurements can also be used to determine the effectiveness of emission control measures (Mijling et al., 2009; Witte et al., 2009; Wu et al., 2013; Chan et al., 2015). However, the uncertainties of satellite trace gas column retrieval are strongly dependent on the accuracy of the assumptions of trace gas vertical distributions. In addition, the

temporal resolution of satellite measurements is often limited to single observation per day. Therefore, it is useful to compare and integrate ground based and satellite observations for the interpretation of the spatial and temporal variation of $NO_2$ and HCHO.

Nanjing is the second largest city in the Yangtze River Delta and Eastern China. It is also the provincial capital of Jiangsu province. The population of Nanjing is about 8 million. Yangtze River is running through the city of Nanjing making it the largest inland port in China. Industrial, water and road transportation are the major anthropogenic sources of air pollutions in Nanjing. Due to its rapid development as well as its surrounding cities in the Yangtze River Delta, Nanjing is facing a series of air pollution problems in recent years. In addition, Nanjing has also hosted several important international events including the summer Youth Olympic Games in 2014. Therefore, it is important to have a better understanding of the air pollution sources in order to support the design of air quality related environmental policies in the future.

In this paper, we present long term MAX-DOAS observations of $NO_2$ and HCHO in Nanjing. Ground based MAX-DOAS measurements were performed from April 2013 to February 2017. Details of the MAX-DOAS experimental setup, the spectral analysis as well as the retrieval of the aerosol extinction, $NO_2$ and HCHO profiles are presented in section 2. Section 3.1 shows the validation of MAX-DOAS aerosol observations by comparing the reported aerosol optical depths (AODs) to sun photometer measurements. The comparison of $NO_2$ and HCHO VCDs measured by the MAX-DOAS and OMI satellite is presented in Section 3.2. An analysis of regional transport of pollutants over Yangtze River Delta is shown in section 3.4. In section 3.5, we presented an evaluation of the pollution reduction observed during the Youth Olympic Games in 2014.

## 2 Methodology

### 2.1 MAX-DOAS measurements

#### 2.1.1 Experimental setup

A MAX-DOAS instrument was set up at a meteorological station of Nanjing University (32.12°N, 118.95°E) which is located on a small hill in the University campus at about 45 m above sea level. The meteorological station is located about 17 km northeast of the Nanjing city center and about 5 km south of the Yangtze River. The MAX-DOAS instrument for scattered sun-light measurements consists of a scanning telescope, a stepping motor controlling the viewing zenith angle of the telescope and a spectrometer. Scattered sun-light collected by the telescope is redirected by a quartz fiber to the spectrometer for spectral analysis. The field of view of the telescope is about 0.6°. An Ocean Optic USB2000 spectrometer equipped with a Sony ILX511 charge-coupled device (CCD) detector is used to cover the wavelength range from 288 nn to 434 nm. The full width half maximum (FWHM) spectral resolution of the spectrometer is 0.6 nm (at 360 nm).

A complete measurement cycle consists of scattered sun-light observations at elevation angle ($\alpha$) of 1°, 2°, 3°, 6°, 10°, 18°, 30° and the zenith (90°). The viewing azimuth angle is adjusted to 320° (northwest). The exposure time and the number of scan of each measurement is adjusted automatically depending on the received intensity of the scattered sun-light spectrum in order to achieve a similar intensity level for all the measurements. A full measurement sequence takes about 5 - 10 minutes.

### 2.1.2 Spectral retrieval

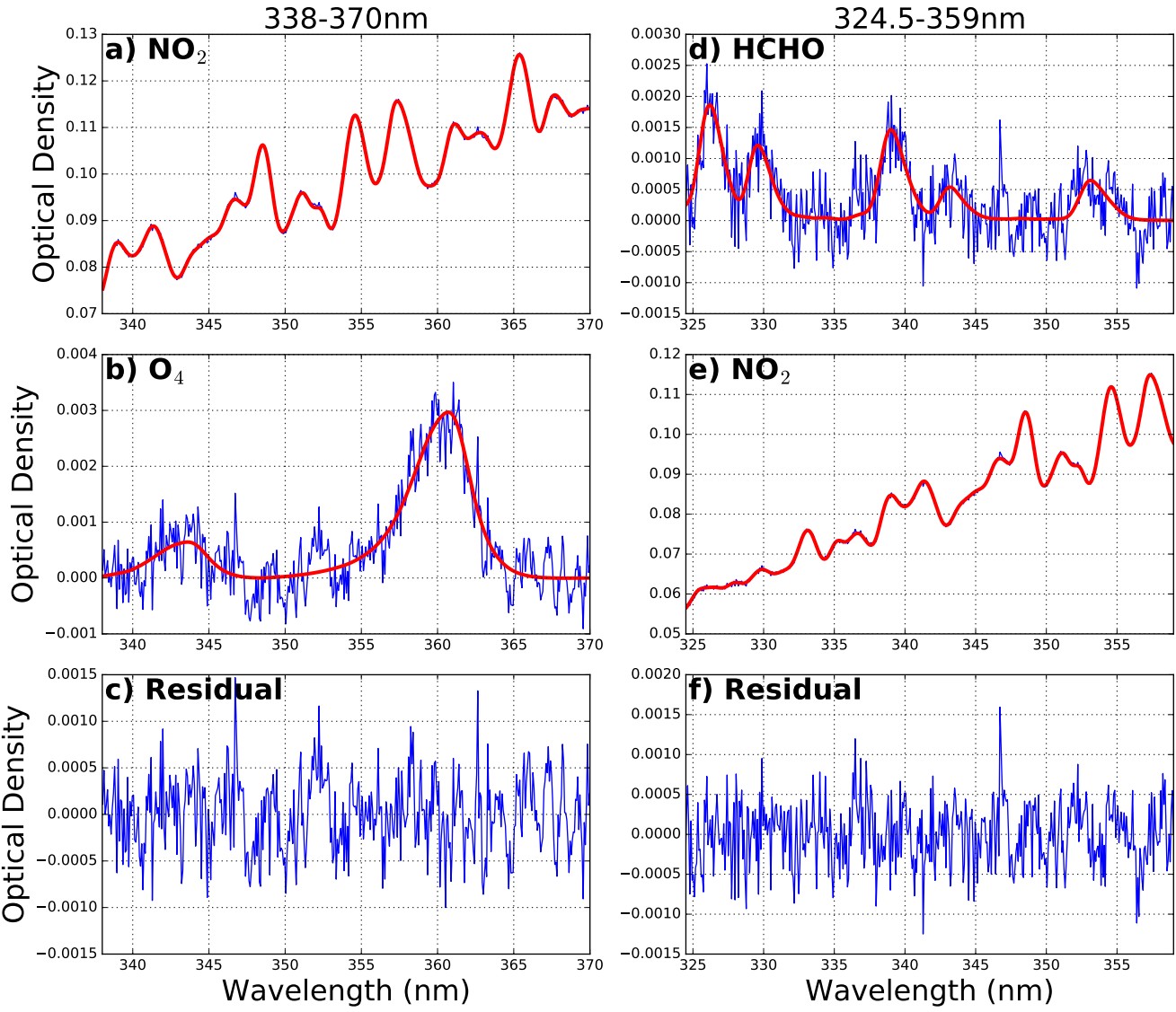

**Figure 1.** An example of the DOAS retrieval of $NO_2$ and HCHO DSCDs from a MAX-DOAS spectrum taken on $12^{th}$ May 2013 at 08:48 (local time) with viewing elevation angle of $1°$. The left panels show the DOAS fit in the wavelength range of $338$ - $370$ nm, while the right panels show DOAS fit in the wavelength range of $324.5$ - $359$ nm.

All the measurement spectra were first corrected for offset and dark current and then analyzed using the spectral analyzing software QDOAS version 3.2 (http://uv-vis.aeronomie.be/software/QDOAS/). The MAX-DOAS spectral fit was performed at

2 different wavelength ranges of 338 - 370 nm for $NO_2$ retrieval and 324.5 - 359 nm for HCHO retrieval. In this study, the zenith spectrum ($\alpha = 90°$) taken in the same measurement cycle is used as reference spectrum in the spectral analysis to retrieve the differential slant column densities (DSCDs). The differential slant column density (DSCD) is defined as the difference between the slant column density (SCD) of the measured spectrum and the corresponding zenith reference spectrum. The broad band spectral structures caused by Rayleigh and Mie scattering are removed by including a low order polynomial in the DOAS fit. Absorption cross section of several trace gases were included in the DOAS fit at different fitting ranges, details of the DOAS retrieval settings for each wavelength range are listed in Table 1. The DOAS fit settings follow the one used in the Quality Assurance for Essential Climate Variables (QA4ECV) project (http://www.qa4ecv.eu/). These settings have also been adopted for the Second Cabauw Intercomparison campaign for Nitrogen Dioxide measuring Instruments (CINDI-2) campaign (Kreher et al., 2019). Small shift and squeeze of the wavelengths are allowed in the wavelength mapping process in order to compensate small uncertainties caused by the instability of the spectrograph. Figure 1 shows an example of the spectral fit retrieval of $NO_2$ and HCHO DSCDs from a MAX-DOAS spectrum taken on $12^{th}$ May 2013 at 08:48 (local time) with viewing elevation angle of $1°$.

**Table 1.** The DOAS retrieval settings for different wavelength bands.

| Species | Temperature | Wavelength Range | | Reference |
| | | 324.5 - 359 nm | 338 - 370 nm | |
| --- | --- | --- | --- | --- |
| BrO | 223 K | ✓ | ✓ | Fleischmann et al. (2003) |
| HCHO | 298 K | ✓ | ✓ | Meller and Moortgat (2000) |
| $NO_2^{[a]}$ | 298 K | ✓ | ✓ | Vandaele et al. (1998) |
| $NO_2^{[a]}$ | 220 K | | ✓ | Vandaele et al. (1998) |
| $O_3^{[b]}$ | 223 K | ✓ | ✓ | Serdyuchenko et al. (2014) |
| $O_3^{[b]}$ | 243 K | ✓ | ✓ | Serdyuchenko et al. (2014) |
| $O_4$ | 293 K | ✓ | ✓ | Thalman and Volkamer (2013) |
| Ring | | ✓ | ✓ | |
| Polynomial | | $5^{th}$ order | $5^{th}$ order | |

[a] $I_0$ correction is applied with SCD of $10^{17}$ molec/cm$^2$ (Aliwell et al., 2002).

[b] $I_0$ correction is applied with SCD of $10^{20}$ molec/cm$^2$ (Aliwell et al., 2002).

As clouds are not included in the radiative transfer calculation of the aerosol and trace gas profile retrieval, the retrieval result can be significantly influenced by the present of cloud in the atmosphere. Therefore, the retrieved DSCDs were first filtered by removing cloudy scenes before proceeding to the aerosol and trace gas profile retrieval. The vertical profile of the oxygen collision complex $O_4$ only varies in a small range with atmospheric pressure and temperature, the retrieved $O_4$ DSCDs and (relative) intensities are assumed to vary smoothly with time with the solar and viewing geometry. Rapid change of $O_4$ DSCDs and intensities indicates a sudden change in the radiative transport condition which is likely due to the present of clouds. Therefore, we applied a locally weighted regression smoothing filter (LOWESS) (Cleveland, 1981) with a regression window

of 3 hours to the $O_4$ DSCDs and intensities time series at each elevation angle to filter data influenced by inhomogeneous and/or rapid changes of radiation transport conditions. Data with fast varying $O_4$ DSCDs and intensities were filter out. Only data with slow varying $O_4$ DSCDs and intensities were used in the aerosol and trace gas retrieval. The limitation of this cloud screening algorithm is that the algorithm is not able to distinguish continuous and homogeneous cloud condition. However, it is rare that the cloud does not change for a long time (within an hour) and the cloud layer is homogeneous for all viewing directions.

### 2.1.3 Aerosol and trace gas profile retrieval

Previous studies show that there is a systematic discrepancy between observation and model simulation of $O_4$ DSCDs (Wagner et al., 2009; Clémer et al., 2010; Wagner et al., 2011; Chan et al., 2015; Wang et al., 2016; Chan et al., 2018; Zhang et al., 2018). A scaling correction factor has to be applied to the measured $O_4$ DSCDs in order to bring measurement and model results into agreement. However, the physical meaning of this observation is still not well understood (Ortega et al., 2016; Wagner et al., 2019). Theoretically, the optical path should be the longest under aerosol free condition for off zenith measurement. Thus, the MAX-DOAS measurement of $O_4$ DSCDs should be smaller than the one simulated with pure Rayleigh atmosphere. In this study, we compared the forward simulation of $O_4$ DSCDs under aerosol free condition to the $O_4$ DSCDs retrieved from the MAX-DOAS observations to examine the necessity of $O_4$ correction. Our result shows that the measured $O_4$ DSCDs occasionally exceeded the forward simulation results which implies that correction of $O_4$ DSCDs is necessary. The ratio between simulated and measured $O_4$ DSCD varies from 0.75 to 1.0 for cases where the measured $O_4$ DSCDs exceeded the forward simulation results. Our finding agrees with previous reported scaling correction factors of $O_4$ DSCDs which is ranging from 0.7 up to 1.0. In order to avoid over-correction due to the outliers, we take the $10^{th}$ percentile instead of the minimum value of the simulated and measured $O_4$ DSCD ratio as the correction factor which is $\sim$0.8. All MAX-DOAS observations of $O_4$ DSCD are corrected by multiplying the correction factor of 0.8. From hereafter, all $O_4$ DSCDs are referring to the corrected $O_4$ DSCDs.

The conversion of the MAX-DOAS observations to aerosol extinction and trace gas profile requires inversion of the underlying radiative transfer equations. These equations cannot be linearized, therefore, it is suggested to fit the measurement quantities to the forward calculation of radiation transfer (Wagner et al., 2004; Hönninger et al., 2004; Sinreich et al., 2005; Frieß et al., 2006; Hartl and Wenig, 2013; Chan et al., 2018). As the vertical distribution profile of $O_4$ is very stable and it has several absorption bands in the ultraviolet and visible spectral range, it is commonly used as fitting quantity for the aerosols retrieval.

In this study, aerosol vertical profiles are retrieved at the 360 nm $O_4$ absorption band using the Munich Multiple wavelength MAX-DOAS retrieval algorithm ($M^3$) (Chan et al., 2018). The algorithm is developed based on the optimal estimation method (Rodgers, 2000) and utilizes the Library for Radiative Transfer (LibRadTran) model (Emde et al., 2016) as the forward model. In this study, the U.S. standard mid-latitude atmosphere profiles for winter (January) and summer (July) are temporally interpolated to each month of the year for the radiative transfer simulations. A brief description of the aerosol and trace gases

vertical profile retrieval algorithm and the parameterization used in this study are presented in the following. A more detailed description of the $M^3$ retrieval algorithm can be found in Chan et al. (2018).

In this study, all valid MAX-DOAS observations within a single measurement cycle are grouped together for the aerosol vertical profile retrieval. Assuming the set of measurement can be reproduced by the forward model and the forward model results are dependent on the aerosol extinction profile, we can then retrieve the aerosol extinction profile by fitting the forward model results to the MAX-DOAS $O_4$ observations using the iterative Newton-Gauss method. The forward simulations of $O_4$ DSCDs are performed at 360 nm due the strong $O_4$ absorption at this wavelength. As the information contained in the MAX-DOAS observations are most likely not enough to retrieve an unique aerosol extinction profile, we have to supply the necessary information to the aerosol inversion in a form of a priori aerosol profile. As the aerosol load in Nanjing varies in a wide range, using a fix a priori could result in over regularizing the retrieval under higher aerosol load conditions. Therefore, we have implemented an iterative approach to avoid over regularizing the retrieval. We first use a fixed initial a priori to retrieve the aerosol extinction profile. The fixed a priori profile is then scaled to have the same aerosol optical depth retrieved from the previous run. Then the new a priori is used in the next run to retrieve the aerosol extinction profile. This procedure is repeated until the difference between the retrieved and a priori aerosol optical depth is less than 10 % or the number of iteration reaches the limit.

As aerosols are typically emitted and formed close to the surface in urban areas, we assume the a priori aerosol extinction profile follows an exponentially decreasing function with a scale height of 1.0 km. The aerosol optical depth (AOD) of the a priori aerosol profile is set to 0.5 for the retrieval at 360 nm. The uncertainties of the a priori aerosol profile are set to 100 % and the correlation length of the aerosol inversion is assumed to be 0.5 km. As MAX-DOAS measurements are more sensitive to the aerosol and trace gases close to the instrument, therefore, we divided the lowest 3.0 km of the troposphere into 15 layers with thickness of each layer of 200 m. A fix set of single scattering albedo of 0.92, asymmetry parameter of 0.68 and ground albedo of 0.04 is assumed in the radiative transfer calculations. An example of aerosol extinction profile retrieval is indicated in Figure 2a-c.

The aerosol information obtained from the procedure described above were used for the differential box air mass factor ($\Delta$DAMF) calculation for the trace gas profile inversion. The $\Delta$DAMFs were calculated at a single wavelength for the retrieval of trace gas profile using the radiative transfer model LibRadTran with the Monte Carlo simulation module MYSTIC (Emde et al., 2016). The $\Delta$DAMF was assumed to be constant within the DOAS spectral fitting windows. As $NO_2$ DSCDs were retrieved in the same fitting window as $O_4$, therefore, we adapted the same wavelength of 360 nm for $NO_2$ $\Delta$AMF simulations. An example of $NO_2$ profile retrieval is indicated in Figure 2d-f. HCHO vertical distribution profile retrieval were retrieved at 340 nm which is close to the center wavelength of the DOAS fitting window of 342 nm. In addition, this wavelength is commonly used in HCHO retrieval, e.g., De Smedt et al. (2018). Aerosol extinction profiles obtained at the 360 nm $O_4$ band are converted to 340 nm assuming a fixed Ångström coefficient (Ångström, 1929) of 1 for the calculation of $\Delta$AMF. The Ångström coefficient can varies significantly with time. Sun photometer measurements in Nanjing (see Section 2.2) show that the Ångström coefficient mostly varies between 0.7 ($10^{th}$ percentile) and 1.4 ($90^{th}$ percentile). Based on the sun photometer observations, we estimated the error caused by fixed Ångström coefficient on aerosol extinction is less than 2 %. The $\Delta$DAMFs

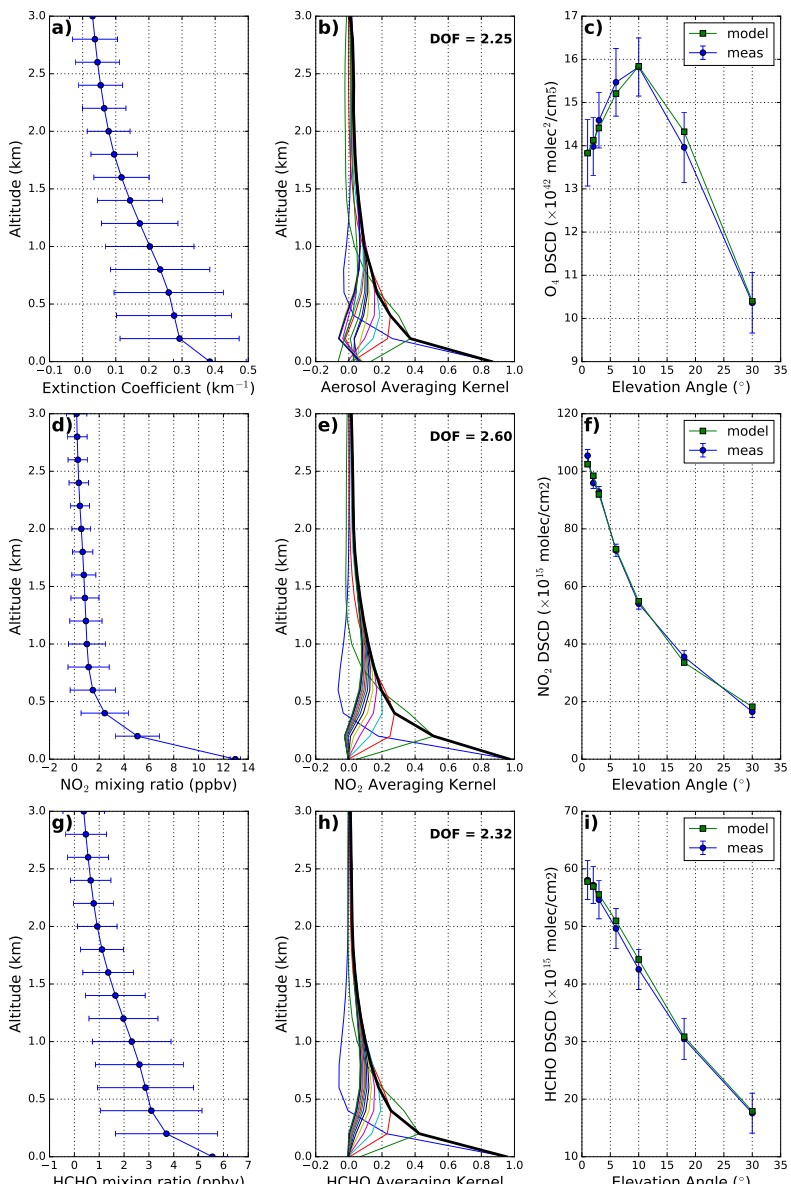

**Figure 2.** An example of vertical profile retrieval of aerosol extinction, NO$_2$ and HCHO from MAX-DOAS measurements taken on $4^{th}$ July 2013 at around 8:47 (local time). The left panels show the (a) aerosol extinction, (d) NO$_2$ and (g) HCHO profiles. Panels in the middle column indicate the average kernel of (b) aerosol extinction, (e) NO$_2$ and (h) HCHO profile retrieval. The measured (blue curves) and modeled (green curves) DSCDs of (c) O$_4$, (f) NO$_2$ and (i) HCHO are shown in the right column.

are subsequently calculated at 340 nm for the HCHO profile retrieval. An example of HCHO profile retrieval is indicated in Figure 2g-i.

The atmosphere layer settings of the trace gas profile retrieval are the same as the one used in the aerosol profile retrieval. $NO_2$ and HCHO concentrations above the retrieval height (3 km) are assumed to follow the U.S. standard atmosphere (Anderson et al., 1986). As the trace gas profile cannot be fully reconstructed by the small number of measurements, therefore, we use the optimal estimation method (Rodgers, 2000) with iterative approach to regularize the inversion (Chan et al., 2015, 2018).

The first step uses a fixed initial a priori to retrieve the trace gas distribution. The fixed a priori profile is then scaled to have the vertical column retrieved from the first run. The scaled a priori is used in the next run to retrieve the trace gas profile. The process repeats until the difference between the retrieved and a priori trace gas vertical column is less than 10 % or the number of iteration reaches the limit. In this study, the a priori is assumed to follow an exponential decrease function with a scale height of 1.0 km. The uncertainty of the a priori profile is set to 100 % of the a priori and correlation length is set to 0.5 km in the trace

gas profile inversion. The $NO_2$ vertical column density (VCD) of the a priori is set to $2 \times 10^{16}$ molec/cm$^2$ while the a priori HCHO VCD is set to $1 \times 10^{16}$ molec/cm$^2$.

## 2.2 Sun photometer measurements

A sun photometer was installed on the roof of a building of Nanjing University of Information Science and Technology (32.20°N, 118.70°E), which is located at the north bank of the Yangtze River. The sun photometer site is ~25 km northwest

of the MAX-DOAS measurement site. The two instruments are separated by the Yangtze River. An industrial park is about 3 km east of the sun photometer site. Heavy industry factories, i.e., steel and cement plants are located in the industrial park. In this study, AODs measured by the sun photometer were used for the inter-comparison study to validate the MAX-DOAS aerosol retrieval. Cloud screened data were used. The data consist of AOD measurements at 7 different wavelength channels from 340 nm to 1020 nm. AOD measurements at 340 nm and 380 nm are interpolated to 360 nm for comparison.

## 2.3 Meteorological data

Meteorological parameters such as temperature, wind speed and wind direction are taken from a weather station in Nanjing which is located about 20 km south of the MAX-DOAS measurement site. The weather station is operated by the National Meteorological Center of China Meteorological Administration. These data are available on the webpage of China Meteorological Administration (https://data.cma.cn/site/index.html). The meteorological data are mainly used for the analysis of

25 meteorological effects on air quality in Nanjing during the Youth Olympic Games in 2014.

## 2.4 OMI satellite observations

The Ozone Monitoring Instrument (OMI) is a passive nadir-viewing satellite borne imaging spectrometer (Levelt et al., 2006) on board the Earth Observing System's (EOS) Aura satellite. The EOS Aura satellite orbit at an altitude of ~710 km with a local equator overpass time of 13:45 LT (local time). The instrument consists of two charge-coupled devices (CCDs) covering

a wavelength range from 264 nm to 504 nm. A scan provides measurements at 60 positions across the orbital track covering a

swath of approximately 2600 km. The spatial resolution of OMI varies from $\sim$320 km$^2$ (at nadir) to $\sim$6400 km$^2$ (at both edges of the swath). The instrument scans along 14.5 sun-synchronous polar orbits per day providing daily global coverage.

The NASA's OMI NO$_2$ and HCHO standard product version 3 (Krotkov et al., 2017; González Abad et al., 2015) are used in this study. In the NO$_2$ product, the slant column densities (SCDs) of NO$_2$ are derived from Earth's reflected spectra in the visible range (402 - 465 nm) using an iterative sequential algorithm (Marchenko et al., 2015). Previous studies show the updated SCDs are on average 10 - 40 % lower compared to the previous version of retrieval (Marchenko et al., 2015). The OMI NO$_2$ SCDs are converted to VCDs by using the concept of air mass factor (AMF) (Solomon et al., 1987). The AMFs are calculated using NO$_2$ profile simulated by the Global Modeling Initiative (GMI) chemistry transport model. The horizontal resolution of GMI is 1° (latitude) $\times$ 1.25° (longitude) (Rotman et al., 2001). Separation of stratospheric and tropospheric columns is achieved by the local analysis of the stratospheric field over unpolluted areas (Bucsela et al., 2013).

The OMI HCHO retrieval algorithm uses the direct fit of radiances in the spectral range from 328.5 nm to 356.5 nm for the SCD retrieval. In the current version of HCHO product, OMI radiance measurement over the remote Pacific is used as reference in the fitting process. This approach is reported to reduce the interferences from unresolved spectral structures in the retrieval of weak absorbers like HCHO (De Smedt et al., 2018). The retrieved SCDs are then converted to VCDs using the AMF approach. The AMFs are calculated based on climatological HCHO profiles.

## 2.5  Back trajectory modeling

The National Oceanic and Atmospheric Administration Air Resources Laboratory (NOAA ARL) developed HYbrid Single Particle Lagrangian Integrated Trajectory (HYSPLIT) model (Stein et al., 2015) (http://www.arl.noaa.gov/HYSPLIT.php) was used to investigate the transportation pollutants over Yangtze River Delta. Meteorological data from the Global Data Assimilation System (GDAS) with a spatial resolution of 0.5° $\times$ 0.5° and 24 vertical levels was used in the model for trajectory simulations. Backward trajectories are computed for air masses arriving at the mid point of each MAX-DOAS retrieval layer.

## 3  Results and discussions

### 3.1  Compariosn of MAX-DOAS and sun photometer AODs

Aerosol optical depths (AODs) retrieved from the MAX-DOAS observations are compared to the sun photometer measurements. As the sampling resolution of the MAX-DOAS and the sun photometer are different, individual measurements are averaged to hourly, daily and monthly values for comparison. Figure 3 shows the scatter plot of AOD measured by the sun photometer and MAX-DOAS. Both datasets are cloud filtered. AODs measured by both MAX-DOAS and sun photometer are in general in good agreement. The Pearson correlation coefficient ($R$) of daily averaged data is 0.73. The slope of the total least squares regression between the two datasets is 0.56 with an offset of 0.13. The discrepancies between the two datasets can be explain by the difference of measurement technique and measurement location. The sun photometer derives AOD from direct sun measurements, while the MAX-DOAS retrieves AOD using the O$_4$ absorption information from scattered sun light.

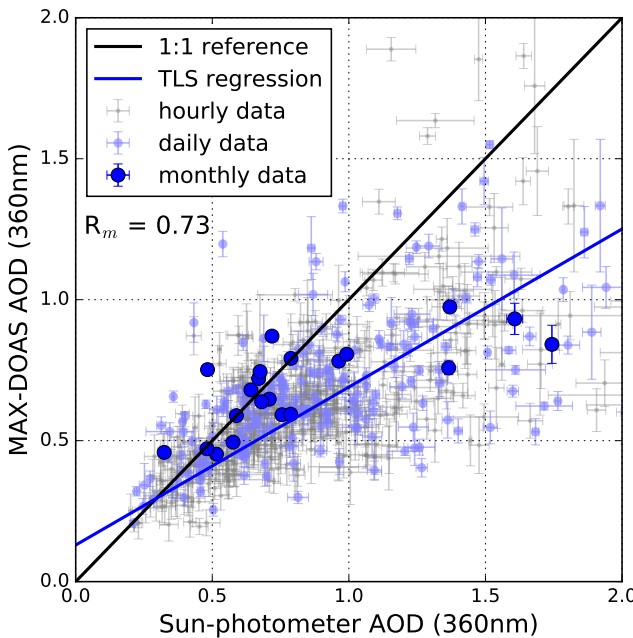

**Figure 3.** Comparison of AOD measured by the MAX-DOAS and sunphotometer. The total least squares regression line is calculated based on the daily data.

The sun photometer is sensitive to the entire column while the MAX-DOAS is mostly sensitive to aerosol at the lowest few kilometers of the troposphere. Therefore, the MAX-DOAS is likely to underestimate the AOD when there is an elevated aerosol layer in the upper troposphere. This explanation matches with the observations that the MAX-DOAS agrees better with the sun photometer under low aerosol conditions (AOD < 1) and underestimate the AOD during high AOD conditions. Another source of error is the assumption of aerosol optical properties in the MAX-DOAS retrieval. In order to quantify the error caused by the fixed asymmetry parameter and single scatter albedo used in the aerosol retrieval, we have performed a sensitivity analysis by retrieving the aerosol profile with different asymmetry parameter and single scattering albedo. As the sun photometer in Nanjing does not provide full inversion of aerosol optical properties, aerosol optical properties are taken from the Taihu AERONET station which is about 150 km southeast of Nanjing. The $10^{th}$ and $90^{th}$ percentile of asymmetry parameter and single scatter albedo are used for the sensitivity study. The result shows that the error caused by the fixed asymmetry parameter on AOD retrieval can be up to 4 %, which the single scattering albedo shows a smaller impact of up to 1.5 %. In addition, the MAX-DOAS and sun photometer are separated by a distance of ∼25 km and the sun photometer site is very close to an industrial park, where the heavy polluted industries are located, i.e., steel and cement plants. Therefore, the sun photometer measurements are expect to be higher than the MAX-DOAS observations. Accounting for these effects, the MAX-DOAS and

sun photometer observation agrees well with each other and the MAX-DOAS derived aerosol profiles are reliable for the retrieval of NO$_2$ and HCHO profiles.

## 3.2 Comparison of MAX-DOAS and OMI observations

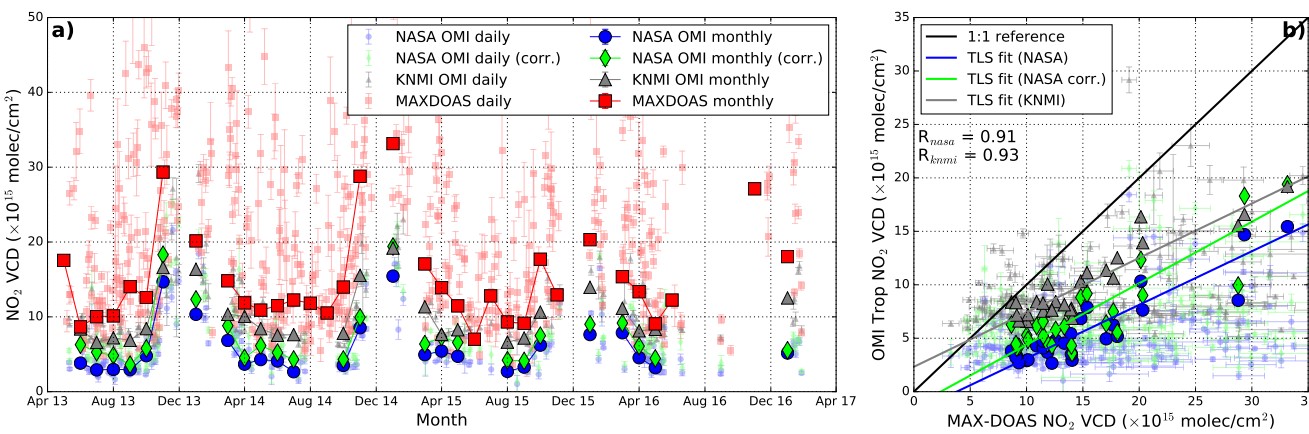

**Figure 4.** (a) Time series of the MAX-DOAS (red markers) and NASA's OMI (blue markers) tropospheric NO$_2$ VCDs over Nanjing from 2013 to 2017. MAX-DOAS data are temporally averaged around the OMI overpass time. OMI measurements are spatially averaged for pixels within 20 km of the MAX-DOAS measurement site. OMI NO$_2$ VCDs retrieved using MAX-DOAS profile as a priori information are indicated as green markers. Tropospheric NO$_2$ VCDs taken from KNMI product are also shown as gray marker for reference. (b) Comparison of tropospheric NO$_2$ VCDs between the MAX-DOAS and OMI satellite observations. The total least squares regression line is calculated based on the monthly average data.

Vertical column densities of NO$_2$ and HCHO retrieved from the ground based MAX-DOAS measurements are compared to OMI satellite observations. MAX-DOAS data are temporally averaged around the OMI overpass time from 12:00 to 14:00 (local time), while OMI VCDs are spatially averaged for pixels within 20 km of the MAX-DOAS measurement site. Time series of MAX-DOAS and OMI observations of NO$_2$ and HCHO VCDs are shown in Figure 4 and Figure 5, respectively. Both OMI NO$_2$ and HCHO datasets are filtered for cloud radiance fraction smaller than 0.4. Daily and monthly averaged data are shown. Missing data for some months are due to cloud filtering or instrumental issues of the MAX-DOAS.

NO$_2$ VCDs measured by the MAX-DOAS and the OMI satellite shows a similar seasonal pattern with higher NO$_2$ columns during winter and lower NO$_2$ values over summer. Higher NO$_2$ levels are due longer atmospheric lifetime of NO$_2$ during winter and higher emissions in winter, e.g., higher power consumption and emissions from individual domestic heating. MAX-DOAS and OMI observations show good temporal consistency with each other with Pearson correlation coefficient ($R$) of 0.91. Despite the strong correlation between the two datasets, the OMI observations is systematically underestimating the NO$_2$ columns. The slope and offset of the total least squares regression between the two datasets is 0.50 and -1.89 $\times$ 10$^{15}$ molec/cm$^2$, respectively. Averaged OMI NO$_2$ column over Nanjing is 5.67 $\times$ 10$^{15}$ molec/cm$^2$ which is about 60 % lower than the MAX-

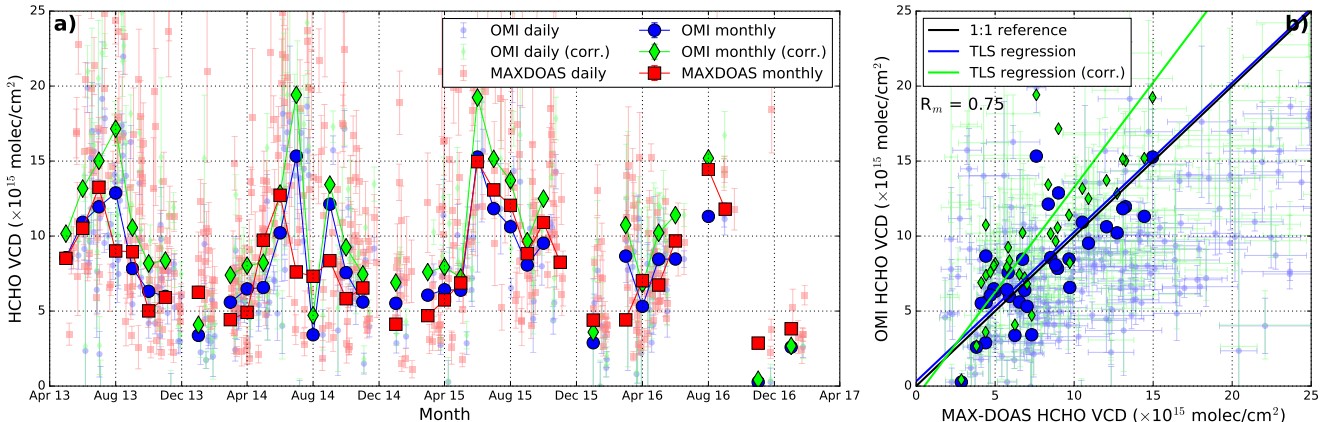

**Figure 5.** (a) Time series of the MAX-DOAS (red markers) and OMI (blue markers) HCHO VCDs over Nanjing from 2013 to 2017. MAX-DOAS data are temporally averaged around the OMI overpass time. OMI measurements are spatially averaged for pixels within 20 km of the MAX-DOAS measurement site. OMI $NO_2$ VCDs retrieved using MAX-DOAS profile as a priori information are indicated as green markers. (b) Comparison of HCHO VCDs between the MAX-DOAS and OMI satellite observations. The total least squares regression line is calculated based on the monthly average data.

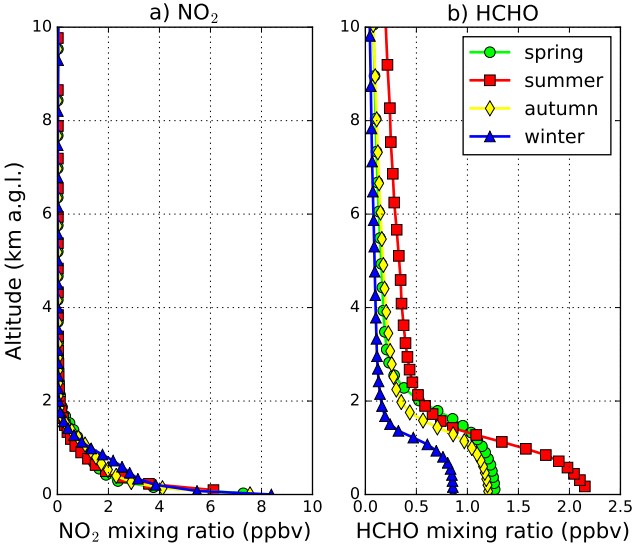

**Figure 6.** Seasonal average of (a) $NO_2$ and (b) HCHO a priori profile used in OMI retrieval. A priori profile of spring (March, April, May), summer (June, July and August), autumn (September, October, November) and winter (January, February and December) are shown.

DOAS averaged VCD of $14.96 \times 10^{15}$ molec/cm$^2$ (see Figure 4). The underestimation of NO$_2$ VCDs from OMI measurements agree with previous studies in the Yangtze Delta (Chan et al., 2015). The discrepancies can be related to the differences in spatial coverage between the ground based and satellite observations and the uncertainties related to the NO$_2$ vertical distribution profile shape used in the air mass factor calculation. We quantified the influence of NO$_2$ vertical distribution profile on the

air mass factor calculation by recomputed the OMI tropospheric NO$_2$ air mass factor using the MAX-DOAS NO$_2$ profile as a priori information. Seasonal average of OMI a priori NO$_2$ profiles are shown in Figure 6a. MAX-DOAS observations of NO$_2$ profiles are shown in Section 3.3. The MAX-DOAS observations show that most of the NO$_2$ is resided below 500 m, while OMI a priori NO$_2$ profiles show a larger fraction of NO$_2$ above 500 m. OMI NO$_2$ VCDs retrieved with MAX-DOAS NO$_2$ profiles are also indicated in Figure 4. Using the MAX-DOAS profile as a priori information in the OMI NO$_2$ VCDs retrieval

on average increased the OMI NO$_2$ VCDs by $\sim$30 % with correlation nearly unchanged. The averaged OMI NO$_2$ column over Nanjing increased to $7.29 \times 10^{15}$ molec/cm$^2$. In addition, the slope of the regression line increases from 0.50 to 0.57 while the offset reduces from -1.89 to $-1.35 \times 10^{15}$ molec/cm$^2$. The result indicates that a better estimation of NO$_2$ vertical distribution in the satellite VCD retrieval reduces the discrepancies. Our findings agree with previous studies that using better estimated NO$_2$ vertical profile in the satellite air mass factor calculation enhanced the OMI NO$_2$ columns and reduced discrepancies between

OMI and ground based observations (Chan et al., 2012; Lin et al., 2014; Chan et al., 2015; Wang et al., 2017). However, the improved OMI NO$_2$ VCDs are still about 50 % lower than the MAX-DOAS observations indicating that there are still remaining issues. Previous study shows that the OMI NO$_2$ tropospheric vertical columns processed by the Royal Netherlands Meteorological Institute (KNMI) (DOMINO version 2.0) (Boersma et al., 2011) only underestimated the tropospheric NO$_2$ VCDs by $\sim$20 % compared to ground based MAX-DOAS observations in Wuxi (another city in Yangtze River Delta) (Wang

et al., 2017). In order to investigate the differences between different OMI NO$_2$ products, we have also plotted the KNMI OMI NO$_2$ product in Figure 4. The average NO$_2$ column reported from the KMI product is $10.10 \times 10^{15}$ molec/cm$^2$, which is about a factor of 2 higher than the NASA product. The result shows the KNMI product only underestimate the NO$_2$ columns in Nanjing by $\sim$30 % which is consistent with previous study in Yangtze River Delta (Wang et al., 2017). The difference between the two OMI products can be related to different spectral analysis techniques, stratospheric tropospheric separation algorithms

as well as a priori profiles used in the retrievals. Investigation of the differences between the two algorithms is however beyond the scope of this study. Another source of discrepancy is related to the difference in spatial coverage of OMI and MAX-DOAS observations. The spatial coverage of the OMI measurement footprint is ranging from 330 km$^2$ up to 4600 km$^2$ with an average of 920 km$^2$. Measurements with such large spatial coverage are probably difficult to capture the spatial gradient of NO$_2$ and resulted in an underestimation over pollution hot spots due to the averaging of large OMI footprint. This effect is especially

significant over Nanjing, as it is a local pollution hot spot surrounded by rather clean areas. The underestimation of NO$_2$ columns is coherent with previous measurements over urban pollution hot spots, i.e., Washington DC and Shanghai (Wenig et al., 2008; Chan et al., 2015). In addition, previous measurements over suburban area in Shanghai show better agreement with OMI observations compared to the measurements in the city (Chan et al., 2015) indicating the effect of spatial inhomogeneity of NO$_2$ on the satellite data comparison. However, the impact of this effect is difficult to quantify due to lack of high spatial

resolution data. Cloud contamination could be an important source of error in the satellite retrieval. Previous study show that

the cloud effect on OMI $NO_2$ VCD retrieval is only significant for cloud radiance fraction $> 0.4$ (Wang et al., 2017). The OMI $NO_2$ data used in this study are filtered for cloud radiance fraction smaller than 0.4. Therefore, cloud contamination should only show a negligible effect in the comparison.

We have also compared the MAX-DOAS and OMI observations of HCHO and both datasets show a similar seasonal varia-
tion pattern. The seasonal variation pattern of HCHO is opposite to $NO_2$ with higher HCHO columns during summer and lower in winter. Higher HCHO levels in summer are related to the increases of biogenic emission of precursor VOCs from plants and higher oxidation rate of VOCs. The MAX-DOAS measurements of HCHO VCDs agree well with the OMI observations with Pearson correlation coefficient ($R$) of 0.75. In addition, the absolute value of the columns show a perfect agreement. The average HCHO VCD measured by the MAX-DOAS is $8.04 \times 10^{15}$ molec/cm$^2$ while the averaged OMI HCHO VCD
is $7.89 \times 10^{15}$ molec/cm$^2$. The slope of the total least squares regression between the two datasets is 0.99 with an offset of $0.31 \times 10^{15}$ molec/cm$^2$ (see Figure 4b). In order to quantify the influence of HCHO vertical distribution profile on the air mass factor calculation, we have recomputed the OMI HCHO VCDs using the MAX-DOAS HCHO profiles as a priori information. OMI HCHO VCDs retrieved with MAX-DOAS HCHO profiles are also indicated in Figure 4. Using the MAX-DOAS profile as a priori information in the OMI HCHO VCDs retrieval on average increased the OMI HCHO VCDs by $\sim$25 % with correla-
tion nearly unchanged. The averaged OMI HCHO column over Nanjing increased to $9.95 \times 10^{15}$ molec/cm$^2$. The slope of the regression line also increases from 0.99 to 1.39 while the offset reduces from 0.31 to -0.72 $\times 10^{15}$ molec/cm$^2$. Higher HCHO VCDs retrieved from OMI using MAX-DOAS profiles as a priori is mainly due to ignoring HCHO in the upper altitudes. Seasonal average of OMI a priori HCHO profiles are shown in Figure 6b. The a priori show a considerable amount of HCHO above 3 km especially during summer, while the MAX-DOAS only reports HCHO mixing ratios up to 3 km above ground level.
Sources of HCHO in the troposphere include the oxidation of varies of VOCs, including methane. Some of these VOCs have relatively long atmospheric lifetime, e.g., methane, therefore, they are rather well mixed in the atmospheric and resulting in a larger portion of HCHO in the upper troposphere. The MAX-DOAS could not capture HCHO at higher altitudes. Therefore, using the MAX-DOAS profiles for OMI HCHO retrieval is likely underestimated the air mass factors and overestimated the HCHO total columns.

## 3.3   Seasonal variation of aerosol, $NO_2$ and HCHO vertical profiles

The seasonal variations of pollutant are closely related to meteorological conditions as well as the characteristics of different emission sources. Analyzing the seasonal variation patterns of different atmospheric pollutants can provide further information on the emission source characteristic as well as the atmospheric processes. Figure 7 shows the vertical profiles of $NO_2$ and HCHO for all four seasons. Significant seasonal patterns are observed from the $NO_2$ and HCHO data. The $NO_2$ vertical profiles
show higher $NO_2$ maxing ratios in autumn and winter and lower in spring and summer. $NO_2$ profiles from all seasons show that the $NO_2$ mixing ratios decrease with increasing altitude. The result indicates most of the measured $NO_2$ is produced close to the surface which agrees with the fact that traffic emission is one of the largest source of atmospheric $NO_2$ in Nanjing. HCHO measurements show a different seasonal variation pattern with lower values during winter and higher mixing ratios in summer. The seasonal pattern of HCHO indicates the significant contribution from biogenic emissions from vegetation. Although HCHO

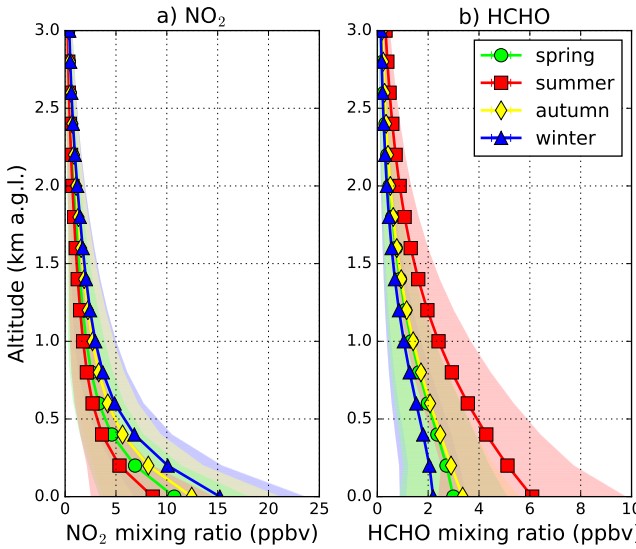

**Figure 7.** Vertical distribution of (a) NO$_2$ and (b) HCHO for different seasons. Shaded areas represent 1 $\sigma$ standard deviation of variation.

profiles for all seasons also show decreasing mixing ratio with height, the decreasing rate is in general much lower than that of NO$_2$. Larger fraction of HCHO is located at higher altitudes. This is probably related to the source characteristic of HCHO. Majority of the tropospheric HCHO is secondary produced from the oxidation of VOCs and resulting in a larger portion of HCHO in the upper altitudes.

## 3.4 Regional pollution transport

Air pollutants released to the atmosphere do not only cause local impacts, but also can influence regions far from the source through regional transport. In order to investigate the influences from regional transportation of air pollutants, we use the backward trajectories calculated by the HYSPLIT model to back correlate the possible source regions of the MAX-DOAS measured NO$_2$. Similar approaches have also been applied for other source appointment studies (Stohl et al., 2009, 2011; Brunner et al., 2012, 2017; Wang et al., 2019). Backward trajectories are calculated at the center point of each MAX-DOAS retrieval layer below 2 km. The MAX-DOAS measurement of NO$_2$ partial columns are then assigned to the grid points along the backward trajectories. We assume the partial columns represent a 200 m thick layer of the assigned grid point and the vertical distribution of NO$_2$ in the assigned grid point is assumed to be homogeneous from the surface to 2 km above ground. In this study, the backward propagated NO$_2$ data are binned in a resolution of $0.2° \times 0.2°$ grid. As NO$_2$ is a relatively short life pollutants in the atmosphere, it is less likely to stay in the atmospheric for a long time and being transported far from the sources. Therefore, NO$_2$ measured by the MAX-DOAS is less correlated to pollutant concentrations far from the site which take longer time to reach the measurement site. In addition, model error accumulates over time. Results with shorter

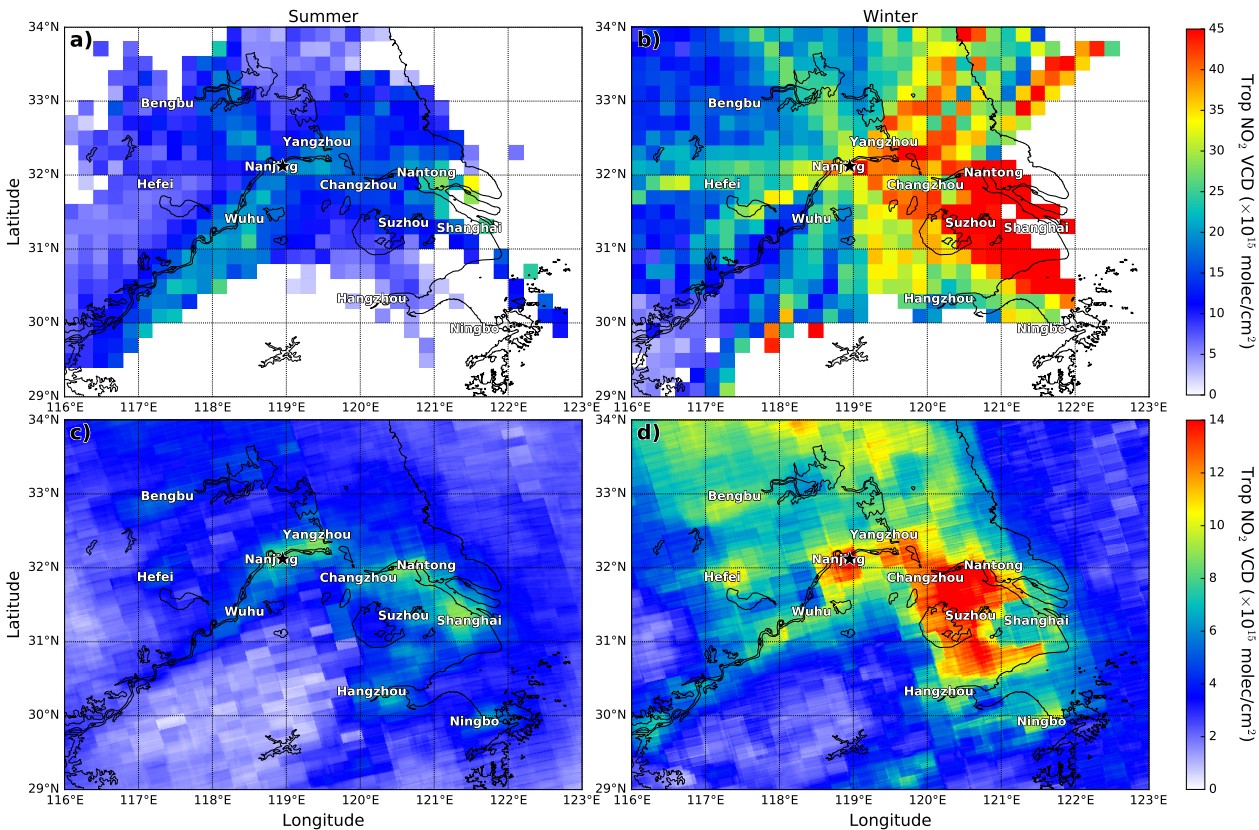

**Figure 8.** The upper panels show spatial distribution of NO$_2$ in (a) summer and (b) winter reconstructed from the MAX-DOAS measurements using backward trajectories calculated by the HYSPLIT model. Backward trajectories are calculated over 6 h for summer measurements and 12 h for winter measurements. The lower panels indicates the average OMI NO$_2$ vertical column densities over the Yangtze River Delta during (c) summer and (d) winter.

simulation time (or backward time) are considered more reliable. In order to account for this effect, an age weighting factor ($w_\tau$) is used in the backward propagation. The age weighting factor is defined by Eq. 1 where $\tau$ is the assumed lifetime of NO$_2$ and $t$ represents the time for the air mass to travel to the measurement site. This weighting scheme is useful when multiple trajectories overlapping with each other within a single grid point.

$$w_\tau = e^{-\frac{t}{\tau}} \tag{1}$$

As the lifetime of NO$_2$ has a strong seasonal variability, we separate the measurements into summer (June, July and August) and winter (December, January and February) in our analysis. Figure 10 shows the spatial distribution of NO$_2$ reconstructed from the MAX-DOAS measurements using different assumed lifetime. The corresponding lifetime used in the calculation and

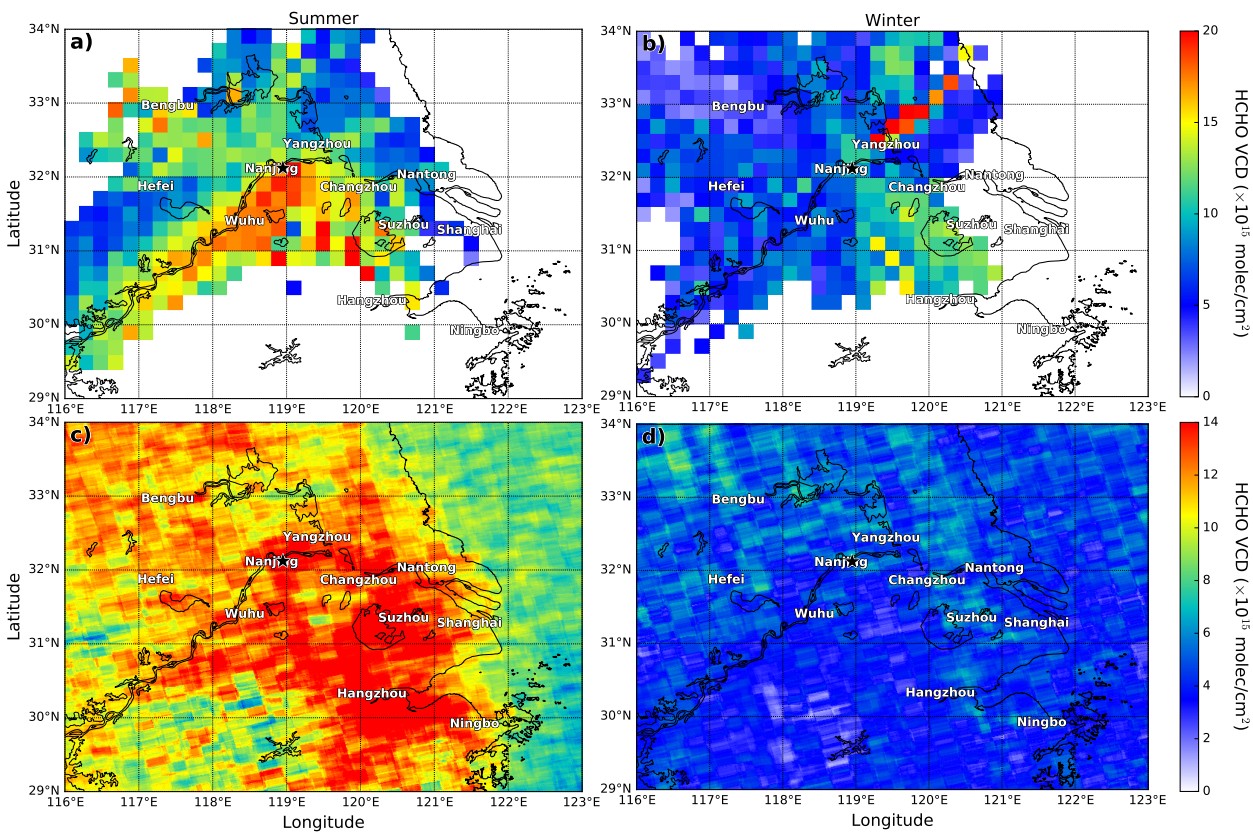

**Figure 9.** The upper panels show spatial distribution of HCHO in (a) summer and (b) winter reconstructed from the MAX-DOAS measurements using backward trajectories calculated by the HYSPLIT model. Backward trajectories are calculated over 6 h for summer measurements and 12 h for winter measurements. The lower panels indicates the average OMI HCHO vertical column densities over the Yangtze River Delta during (c) summer and (d) winter.

the spatial correlation between the reconstructed maps and OMI observations are indicated on the plots. Backward trajectories are calculated over 2 times the assumed lifetime. As the assumed lifetime is only used for the calculation of the age factor, the accuracy of the lifetime does not show a strong impact on the reconstructed spatial pattern. $NO_2$ lifetime of 3 h and 6 h are chosen to reconstruct the spatial distribution of $NO_2$ during summer and winter, respectively. Backward trajectories are calculated over 6 h for summer measurements and 12 h for winter measurements. These values are chosen based on the consideration of getting a balance between having better spatial coverage and the reliability of the reconstructed pollution maps. Note that the simulation time (or backward time) is changing along the trajectory, the weighting of each point along the trajectory is depending on its simulation time according to Eq. 1. Points along the trajectory with shorter simulation time are weighted higher in the calculation. $NO_2$ fields reconstructed from the MAX-DOAS measurements in summer and winter using backward trajectories calculated by the HYSPLIT model are shown in Figure 8a&b. Figure 8c&d shows the averaged OMI

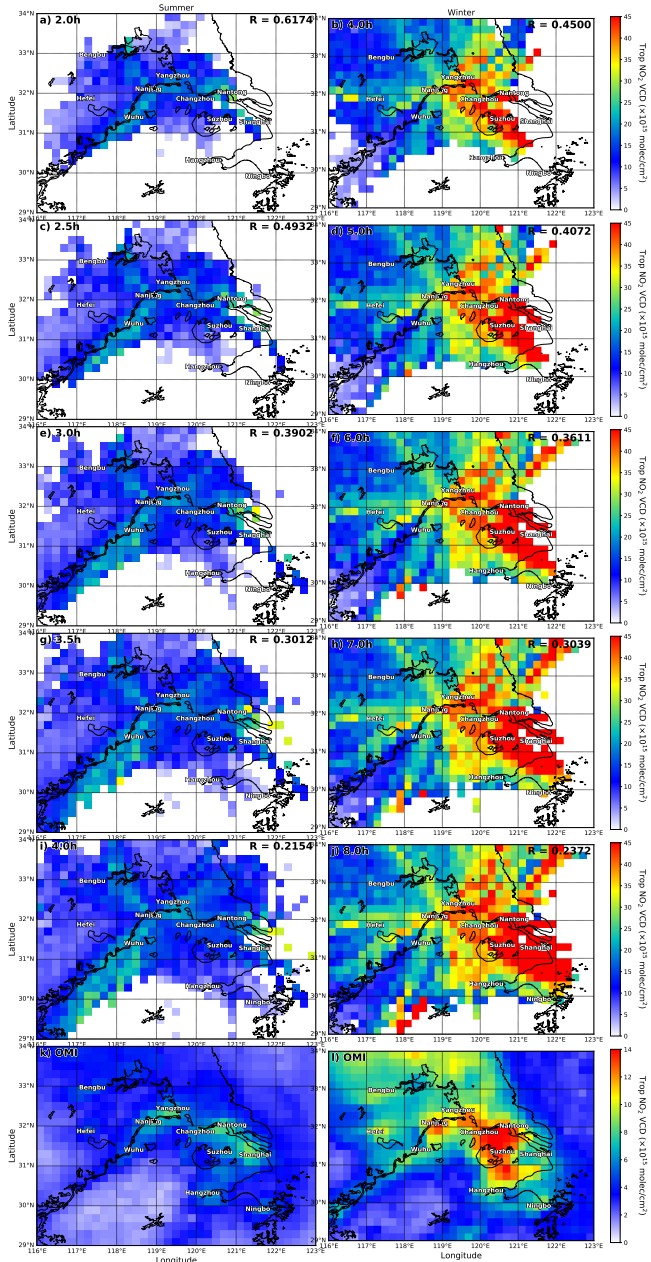

**Figure 10.** Spatial distribution of NO$_2$ in summer (left panels) and winter (right panels) reconstructed from the MAX-DOAS measurements using different assumed lifetime. The average OMI NO$_2$ vertical column densities during summer and winter are shown in (k) and (l), respectively. The corresponding lifetime used in the calculation and the spatial correlation between the reconstructed maps and OMI observations are indicated each plot.

satellite observations of $NO_2$ during summer (June, July and August) and winter (December, January and February) from June 2013 to February 2017. The OMI measurements show that Nanjing has serious $NO_2$ pollution problem in winter.

The reconstructed $NO_2$ fields show a good spatial agreement with OMI satellite observations retrieved by NASA. However, the reconstructed $NO_2$ fields are on average 3 times higher than the OMI observations. Large discrepancies of the absolute values between the reconstructed data and OMI observations are expected, as the differences are also observed in the direct comparison of VCDs measured by the MAX-DOAS and OMI as presented in Section 3.2. Elevated $NO_2$ levels are observed during winter when air mass coming from the east, e.g., Shanghai, Suzhou and Nantong, where OMI observations also show high $NO_2$ levels. The summer time $NO_2$ levels are in general lower. However, elevated values can still be observed along the Yangtze River and over some cities, e.g., Nantong and Wuhu.

We have also applied the same approach to the MAX-DOAS HCHO measurements with HCHO lifetime of 1 h and 2 h for summer and winter measurements, respectively. Due to the considerable contribution of secondary formation of HCHO from its precursors and short atmospheric lifetime of HCHO, relatively short lifetimes are chosen to reconstruction the spatial distribution of HCHO. The reconstructed maps would be more related to the spatial distribution of precursors of HCHO instead of HCHO itself if a longer lifetime is applied in the analysis. HCHO fields reconstructed from the MAX-DOAS measurements are shown in Figure 9a&b. OMI observations of HCHO are also shown in Figure 9c&d for reference. Summer time HCHO data are in general higher than the observations during winter. However, the spatial distribution of the back propagated HCHO data does not show a strong correlation with the OMI satellite observations as the $NO_2$ data. This is probably due to HCHO is mostly secondary produced in the atmosphere. In addition, the atmospheric lifetime of HCHO is much shorter compared to $NO_2$ and therefore more dependent on the local productions. The result suggests that the MAX-DOAS measurements are sensitive to the regional transport of air pollutants. Despite the strong local contributions, the air quality of Nanjing can also be significantly influenced by the air pollution transportation, especially during winter.

### 3.5   Assessments of emission reduction during Youth Olympic

The summer Youth Olympic Games was held from 16 to 28 August 2014 in Nanjing, China. During the event, the government implemented a series of air pollution control measures. Heavy emission vehicles were strictly prohibited. Local heavy industries, e.g., petrochemical and steel industries were required to limit their production during the Youth Olympic Games. These air pollution control measures were often implemented when such an international events held in China (Wu et al., 2013; Wang et al., 2014; Chan et al., 2015; Liu et al., 2016; Sun et al., 2016). In order to study the influences of the emission control measures implemented during the Youth Olympic Games on the local air quality, we compare the MAX-DOAS observations of aerosol, $NO_2$ and HCHO taken before, during and after the Youth Olympic Games. We define the pre-Olympic period from a month before the Youth Olympic Games (16 July 2014) to 1 day before the Youth Olympic Games (15 August 2014). During the pre-Olympic period, the government gradually started to implement some of the emission control measures. During the Youth Olympic Games all those air pollution control measures were strictly implemented. The post-Olympic period is defined from 29 August 2014 to 28 September 2014 where the emission control was gradually getting back to normal.

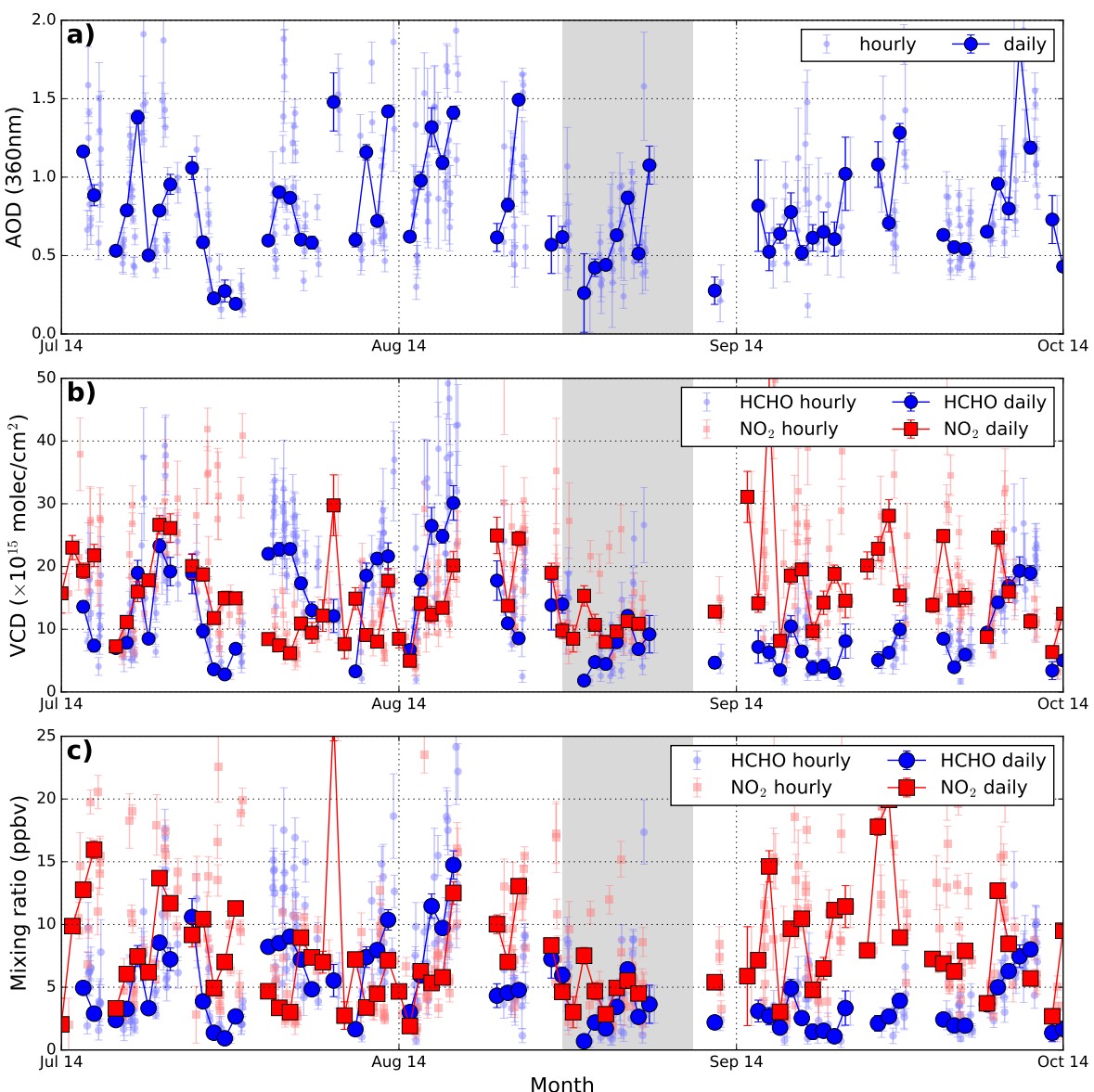

**Figure 11.** Time series of (a) aerosol optical depths, (b) HCHO and $NO_2$ VCDs and (c) HCHO and $NO_2$ surface mixing ratios measured by the MAX-DOAS around the Youth Olympic Games. Hourly and daily averaged values are shown. The gray shadowed area indicates the Youth Olympic Games period (16 to 28 August 2014).

Time series of aerosol optical depths, HCHO and $NO_2$ VCDs and surface mixing ratios measured by the MAX-DOAS around the Youth Olympic Games are shown in Figure 11. The gray shadowed area indicates the Youth Olympic Games period

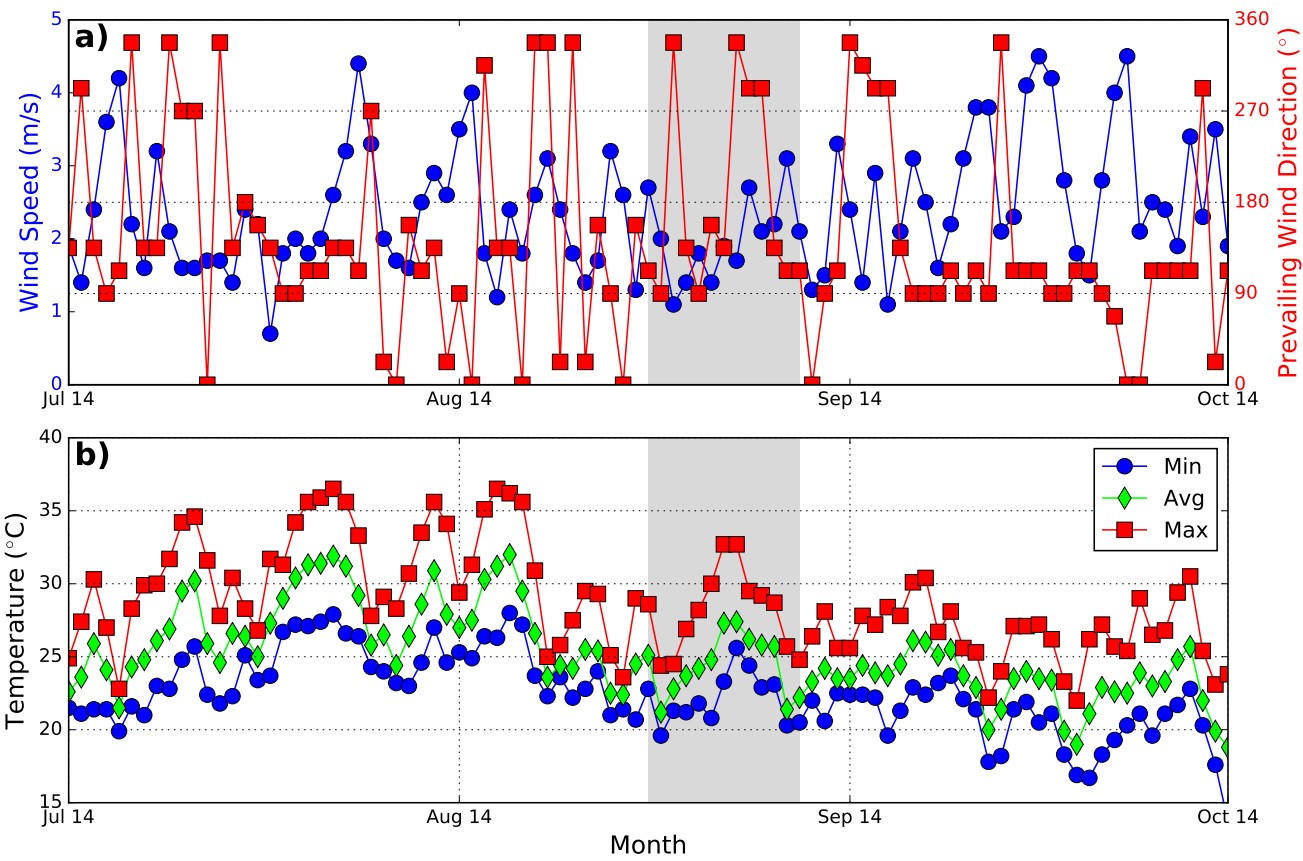

**Figure 12.** (a) Time series of wind speed (blue curve) and wind direction (red curve). (b) Time series of daily minimum (blue curve), daily average (green curve) and daily maximum temperature. The gray shadowed area indicates the Youth Olympic Games period (16 to 28 August 2014).

(16 to 28 August 2014). Meteorological parameters such as temperature, wind speed and wind direction measured by a weather station in Nanjing during the same period are shown in Figure 12. Backward trajectories over the three periods are shown in Figure 13. The backward trajectories show that there are slightly less air masses coming from the southwest direction during the pre-Olympic period, while the Olympic and post-Olympic periods show very similar air mass transportation conditions.

5   The AODs, HCHO and $NO_2$ measurements show a strong temporal variability. In order to investigate the effect of the Youth Olympic Games, we analyzed the statistic of AODs, HCHO and $NO_2$ VCDs for the pre-Olympic, Olympic and post-Olympic periods. Boxplots of the AODs, HCHO and $NO_2$ VCDs for the three periods are shown in Figure 14. The results show that AOD, $NO_2$ and HCHO VCD are the lowest during the Olympic period among those 3 periods. The MAX-DOAS measurements of AOD reduced from 0.9 during the pre-Olympic period to 0.6 for the Olympic period and bounced back to about 0.8 after

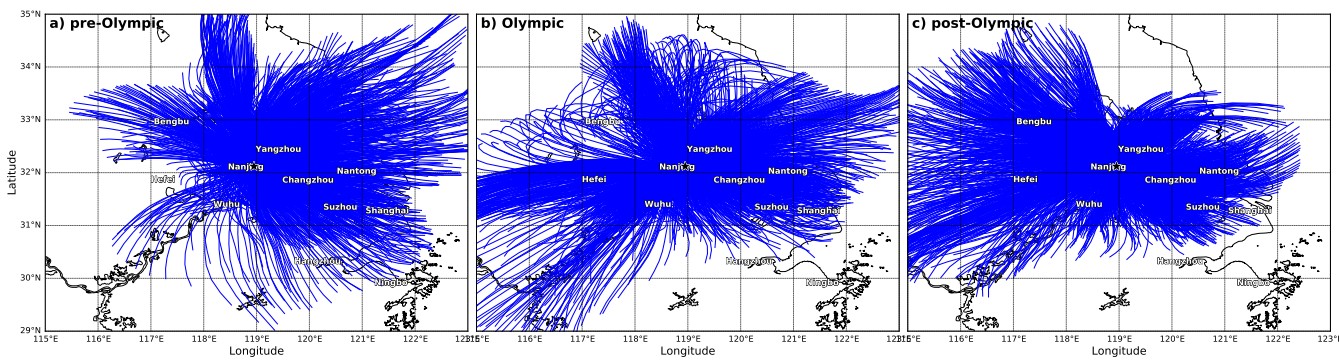

**Figure 13.** Backward trajectories calculated for (a) pre-Olympic, (b) Olympic and (c) post-Olympic periods. Backward trajectories are calculated over 12 h.

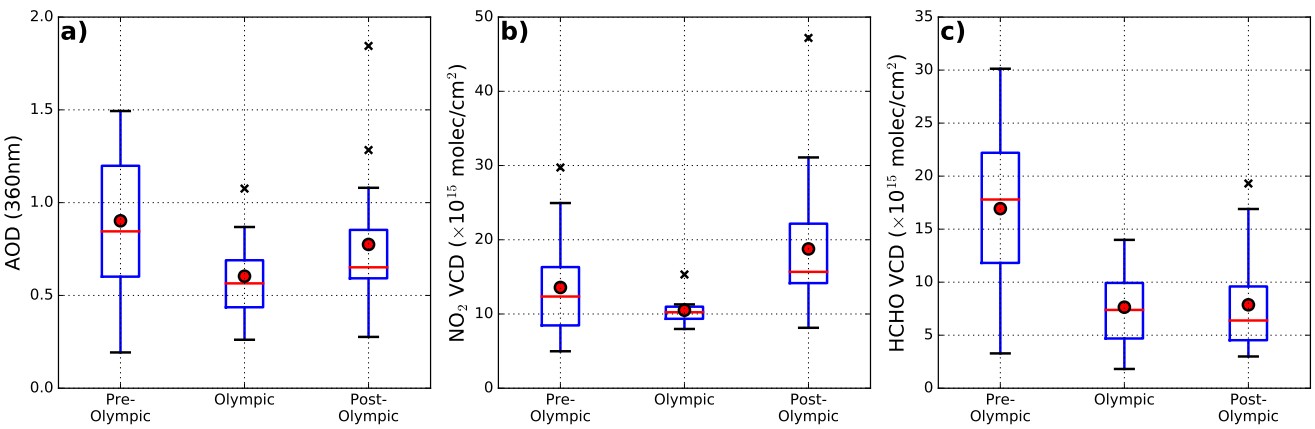

**Figure 14.** Boxplots of (a) aerosol optical depths, (b) $NO_2$ and (c) HCHO VCDs before, during and after the Youth Olympic Games.

the Youth Olympic Games. A reduction of $NO_2$ columns can also be observed during the Youth Olympic Games. Averaged $NO_2$ VCDs are 14, 11 and $19 \times 10^{15}$ molec/cm$^2$ for periods before, during and after the Youth Olympic Games, respectively. Similar reduction pattern is also observed from the $NO_2$ surface mixing ratios. Averaged $NO_2$ surface mixing ratios are 7.5, 4.7 and 9.0 ppbv for periods before, during and after the Youth Olympic Games, respectively. The HCHO columns measured during the Youth Olympic Games also decreased by more than ~50 % from $17 \times 10^{15}$ molec/cm$^2$ before the Olympic down to $8 \times 10^{15}$ molec/cm$^2$ during the Youth Olympic Games. Surface HCHO mixing ratios also show similar reduction. The surface HCHO mixing ratios measured also reduced from 7.0 ppbv before the Youth Olympic to 3.3 ppbv during the Youth Olympic. The emission and secondary formation of HCHO are closely related to the ambient temperature. Higher ambient temperature is usually associated with higher formation rate and natural emission of HCHO and its precursors. The meteorological data

indicates that the ambient temperature is decreasing with time when the season changing from summer to autumn. The HCHO concentrations measured during the Youth Olympic Games are expected to be higher than that measured in the post-Olympic period if anthropogenic emissions are unchanged. However, the measurement shows that the HCHO concentration is slightly lower during the Youth Olympic compared to the post-Olympic period. The result implies that the emissions of HCHO (and its precursors) are reduced during the Youth Olympic Games. As the other meteorological conditions are very similar during the three periods, the decrease of pollutant concentrations are mainly attributed to the reduction of emissions during the Youth Olympic Games.

## 4 Summary and conclusions

In this paper, we present long term observations of atmospheric $NO_2$ and HCHO in Nanjing using a MAX-DOAS instrument. Ground based MAX-DOAS measurements were performed from April 2013 to February 2017. Differential slant columns of $O_4$, $NO_2$ and HCHO were retrieved by applying the differential optical absorption technique to the scattered sun light spectra at the UV band. The results are served as inputs for the retrieval of aerosol extinction, $NO_2$ and HCHO profiles in the lower troposphere.

Aerosol results derived from the MAX-DOAS observations are validated by comparing the AODs to sun photometer observations. The MAX-DOAS and sun photometer measurements show a good agreement with each other with Pearson correlation coefficient ($R$) of 0.73. Considering the differences in measurement technique and measurement location, we concluded that the MAX-DOAS aerosol measurements are reliable for the $NO_2$ and HCHO profile retrieval.

Tropospheric vertical column densities (VCDs) of $NO_2$ derived from MAX-DOAS measurements are used to validate OMI observations. The comparison shows that the OMI observations correlate well with the MAX-DOAS data with Pearson correlation coefficient ($R$) of 0.91. However, OMI observations are on average 60 % lower than the MAX-DOAS measurements. Using the MAX-DOAS $NO_2$ profile as a priori information in the OMI retrieval on average increased the OMI $NO_2$ VCDs by $\sim$30 % with correlation nearly unchanged. However, the improved OMI $NO_2$ VCDs are still about 50 % lower than the MAX-DOAS observations. The KNMI OMI $NO_2$ product shows a better agreement with the MAX-DOAS observations with an underestimation of $\sim$30 %. Large difference between the two OMI product is related to the different spectral analysis techniques, stratospheric tropospheric separation algorithms as well as a priori profiles used in the retrievals. Another source of discrepancy between MAX-DOAS and OMI observations includes the difference in spatial coverage between the two measurements. We also compared the OMI observation of HCHO VCDs to our MAX-DOAS data. The result shows a good agreement between the two datasets with $R$ of 0.75 and the slope of the regression line is 0.99.

The MAX-DOAS measurement of $NO_2$ and HCHO profiles are analyzed together with the backward trajectory simulations to assess the regional transportation and possible source regions of the MAX-DOAS measured $NO_2$ and HCHO. The age weighted backward propagation approach is used to reconstruct the spatial distribution of $NO_2$ and HCHO over the Yangtze River Delta during summer and winter time. The reconstructed $NO_2$ fields show a distinct agreement with OMI satellite observations. The result shows the MAX-DOAS measurements are sensitive to the air pollution transportation in the Yangtze

River Delta. However, due to the short atmospheric lifetime of HCHO, the backward propagated HCHO data does not show a strong spatial correlation with the OMI HCHO observations. Our result suggested the air quality of Nanjing are significantly influenced by air pollution transportation, especially during winter.

We have also used the MAX-DOAS observations of aerosol, $NO_2$ and HCHO for the investigation of the effectiveness of air pollution control measures implemented during the Youth Olympic Games 2014. Our results show a significant reduction (30 % - 50 %) of ambient aerosol, $NO_2$ and HCHO compared to measurements before and after the Youth Olympic games. The results indicate the effects of the reduction of emissions during the Youth Olympic Games. Our findings provide a better understanding of the transportation and sources of pollutants in the Yangtze River Delta as well as the effects of emission control measures during large international event, which are important for the future design of air pollution control policies.

*Author contributions.* KLC, AD, KPH and NH designed the experiment. KLC, ZW, AD, YS and NH carried out the experiment. JW, FZ and YS provided auxiliary data. MW provided useful comments for the discussion. KLC analyzed the measurement data and prepared the manuscript with contributions from all co-authors.

*Competing interests.* The authors declare that they have no conflict of interest.

*Acknowledgements.* The authors would also like to thank the National Oceanic and Atmospheric Administration (NOAA) Air Resources Laboratory (ARL) for the provision of the HYSPLIT transport and dispersion model used in this publication. The work described in this paper was partly supported by the ESA-MOST Dragon 3 Cooperation Programme under the framework of the East Asian monsoon and air quality project (project ID: 10455).

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
