# Peer review of "MAX-DOAS measurements of tropospheric NO2 and HCHO in Nanjing and the comparison to OMI observations"

_Atmospheric Chemistry and Physics, 2018_

## Referee Comment (RC1) · Anonymous Referee #1 · 7 Mar 2019

General comments In this paper, Chan et al. presented the long-term MAX-DOAS measurements of NO2 and HCHO profiles in Nanjing. The data are used to validate OMI NO2 and HCHO products, discuss the effects of a-priori profiles on OMI retrievals, analyze effects of regional transports, and effects of pollution control measures during the Youth Olympic Games. In general the scientific topic is meaningful, and the MAX-DOAS data quality is well proved. However the authors need to give more deep discussions in many parts to firmly prove the conclusions. Generally I have three major concerns below:

1) Regarding the comparisons with the OMI data in section 3.2, the authors should also

[Figure]

show the comparisons of a-priori profile shapes of OMI NO2 retrievals with the NO2 profiles measured by MAX-DOAS for the discussion on the effect of a-priori profiles. In addition, the authors also need to discuss the a-priori effect for HCHO even in the case that a good agreement is found. It is very important to see if good agreements of HCHO profile shapes can been also seen between MAX-DOAS and OMI a-priori.

The underestimation of OMI NO2 VCDs is up to 50% compared to MAX-DOAS data shown in Fig. 2. However the previous study in Wuxi, see Wang et al. 2017, shows the underestimation is ∼20%. One major difference is that the NO2 product is from NASA in your study, but from DOMINO v2 in Wang et al., 2017. Are there big differences of both OMI NO2 data sets? Why are there big differences? In order to answer the questions, the author needs to do comparisons also with the DOMINO v2 product. Meanwhile DOMINO v2 is an official product which is well known and widely used. In addition the author demonstrates that "Measurements with such large spatial coverage are probably difficult to capture the spatial gradient of NO2 and resulted in an underestimation over pollution hot spots due to the averaging of large OMI footprint. This effect is especially significant over Nanjing, as it is a local pollution hot spot surrounded by rather clean areas". If it is true, the NO2 measured by the MAX-DOAS is dominated by local emission. However the discussion on regional pollution transport in section 3.4, the author concludes that "the air quality of Nanjing is significantly influenced by the air pollution transportation, especially during winter.". The two elaborations are contradictive. Therefore the authors need to carefully discuss the reason of the underestimations of the OMI NO2 data.

2) In section 3.4, the authors used the reconstructed maps to quantitatively validate the satellite maps. Therefore a speculated life time is used to scale MAX-DOAS VCD in the reconstruction of maps. However are the quantitative comparisons reasonable? Because the authors assume that all the pollutants measured by the MAX-DOAS instrument are from emissions in an area corresponding to the starting location of a trajectory. However emissions in different grids along the trajectory route should be

mixed up and contribute to the pollutants measured by the MAX-DOAS in reality. The emissions from different distances should be scaled differently. But do we know the proportions of the different emissions? I think the reasonable comparisons of the re-constructed maps with the satellite maps are the relative distributions, but not the ab-solute values.

The reconstructed maps should depend on the selected backward time of trajectories. The author should show the maps with different trajectory backward time and compare them with the satellite maps in order to see which time is reasonable. And the suitable backward time depends on actual lifetime. Since the lifetime effect is already implied if different backward time is tested, the scaling with lifetime might be not needed and do not give any meaningful results. In addition, transports of pollutants can occur during night time and day time. Lifetime only matters for transports during day time. Night time transports can reach a far distance and contribute to concentrations of pollutants during day time. This is another reason why lifetime should not be applied.

The backward propagation method has been applied to long lifetime pollutants and also trace gases measured from MAX-DOAS in previous studies. Some references should be cited in the paper. Meanwhile the sentence "We developed a new technique to assemble the source contribution map using backward trajectory analysis" in the abstract might be inappropriate.

3) In section 3.5, the author compared the pollutants during the Youth Olympic Games with those before and after the event in order to characterize the effect of pollution control measures. Since pollution transports can impact Nanjing as the author demon-strates in section 3.4, the difference of transport conditions in the three periods should also be discussed. Meanwhile the author simply elaborates "As the meteorological conditions are very similar during the three periods". I think the author has to show wind fields, trajectories, precipitations, and temperatures in the three periods in or-der to convince the readers. Near-surface concentrations of the pollutants should be also derived from the MAX-DOAS profile inversion. Since near-surface concentrations

should be mainly dominated by the local emission, but VCDs (AODs) contain contributions of pollutant transports. Therefore it is also meaningful to include the comparisons of the surface concentrations as VCDs shown in Fig.9.

Specific Comments:

1) P4 L3: A reference should be given for QDOAS. Please clarify which of the two wavelength ranges is used for NO2 and HCHO?

2) P4 L4: Please clarify the reference spectrum is the zenith measurement in individual elevation scan or around noon time?

3) Table 1: Do you determine the DOAS fit settings based on sensitivity studies (which are not shown) or previous studies? If you determined them based on previous studies, some references should be given. In addition, do you apply the wavelength dependent Ring suggested by Wagner et al., 2009? If not, please discuss why the additional Ring is not needed in your analysis.

Wagner, T., Beirle, S., and Deutschmann, T.: Three-dimensional simulation of the Ring effect in observations of scattered sun light using Monte Carlo radiative transfer models, Atmos. Meas. Tech., 2, 113-124, 2009.

4) Section 2.1.2: examples of DOAS fits should be shown, especially for HCHO, in order to convince the quality of HCHO analysis.

5) P4 L17-19: How do you filter the data under continuous clouds when the variability of O4 dSCDs are not large?

6) P5 L15: Since O4 VCD can systematically vary during a year due to variations of temperature and pressure, as Wagner et al. (2018 AMT) demonstrated, the phenomenon can explain the scaling factor in many places. How do you consider the variation of temperatures in the retrievals of aerosols? If you don't consider it, a discussion on the uncertainties due to the effect has to be given.

**ACPD**

7) P6 L11: How do you determine the single scattering albedo and asymmetry parameters, and also Ångström coefficien? The parameters can significantly change in the long-term measurements, uncertainty estimations of aerosol results due to the parameters should be given in the paper.

8) P6 L16-18: How do you determine the wavelengths of the AMF simulations of O4, NO2, and HCHO?

9) P6 L20: How do you deal with the NO2 above 3km? The considerable amount of NO2 at high altitudes might also impact retrievals of NO2 below 3km.

10) Section 2.1.3: Figures of comparisons of measured dSCDs and modeled dSCDs for profile retrievals should be shown in the manuscript or supplement to show the convergence of the profile retrievals.

11) Section 2.3: The overpass time of OMI should be given.

12) P8 L13-14: Can the constraint of a-priori profile contribute to the underestimations? In order to show this, comparisons of measured dSCDs and modeled dSCDs are needed.

13) P8 L28: As I know, there are not domestic heating systems in Nanjing since it is in the south of Huai River.

14) P10, L1: The underestimation of OMI NO2 compared to MAX-DOAS is not consistent with Wang et al., 2017. The underestimation here is much stronger.

15) P13, L6: Oxidation rate of VOCs to HCHO is also stronger in summer than in winter. The variations of oxidation rate can also contribute to seasonal pattern of HCHO. And secondary sources of HCHO are significant. The seasonal pattern of HCHO might be due to contributions of biogenic emissions of precursor VOCs. The sentence should be modified.

16) P13, Figure 5: The color scale of subfigure (a) should be changed to allow seeing the gradient more clearly. As I elaborated in General comment (2), the relative distribution is much more important than the absolute values.

17) P13, L16: Since you calculate the trajectories in each altitude grids of MAX-DOAS profiles, how do you combine the trajectories with the profiles? Do you assign partial columns in each vertical grid to different grid points in the map along trajectories at individual altitudes? The procedure need to be clarified.

18) P14, L5-6 how do you determine the lifetime and backward time? The question is corresponding to the general comment (2).

19) P14, L12: As I demonstrate in comment (2), the quantitative comparisons with OMI data are not reasonable.

20) P14, L19: Since HCHO is dominated by the secondary formations from VOCs, which have a long lifetime, therefore VOCs might be transported to a far distance and contribute to local HCHO concentrations. Therefore transport effects on HCHO might be even larger if the transports are from far distance. The backward time of trajectories of 6 hour might be not long enough in the reconstruction of HCHO maps. Following my general comment (2), I suggest you to generate the maps with different backward time of trajectories and compare the relative distributions with OMI maps.

---

## Referee Comment (RC2) · Anonymous Referee #2 · 18 Mar 2019

The manuscript entitled 'MAX-DOAS measurements of tropospheric NO2 and HCHO in Nanjing and the comparison to OMI observations' by Chan et al. presented a long term MAX-DOAS observations of atmospheric nitrogen dioxide (NO2) and formaldehyde (HCHO) in Nanjing. The MAX-DOAS measurements were validated by comparing to sun-photometer observations. The authors then used the MAX-DOAS data for the validation of the NASA's OMI NO2 and HCHO products. OMI observations in general show a good agreement with the ground based observations. A discussion of a priori profile on the satellite retrieval is also presented. The MAX-DOAS data is also used for the investigation of regional transportation of pollutants and for the assessment of air quality during the Youth Olympic Game in Nanjing. The study is in general

well written and scientifically interesting for the community. Therefore, I recommend publishing the manuscript after addressed the following minor comments. Minor comments: 1. Although the agreement between OMI and MAX-DOAS HCHO observations is already very good, it is still interesting to see the effect of MAX-DOAS profiles being used for OMI HCHO VCDs retrieval. I understand that there is a large fraction of HCHO above the MAX-DOAS retrieval height compared to NO2, using the MAX-DOAS profile would result in a larger OMI HCHO columns. This is also relevant for other MAX-DOAS satellite comparison studies. 2. The authors mentioned on page 11 that it is difficult to quantify the effect of spatial inhomogeneity of NO2 on the satellite data comparison due to lack of high spatial resolution data. I am wonder if the MAX-DOAS is still measuring? If yes, then it would be useful to compare the latest measurement to TROPOMI observations. As TROPOMI provides much higher spatial resolution data, these datasets are very useful for the quantification spatial gradient effect on satellite to ground based measurement comparison. 3. The reconstruction of spatial distribution NO2 and HCHO from MAX-DOAS measurements using trajectories simulations are particularly interesting. However, the description of the method is a bit too brief. The author should include a more detail description. 4. In addition, do the authors try other lifetime weight factors in this study? Are the weighting factors determined by fitting to satellite data or the authors just select a realistic one? This can be relevant for other similar studies. 5. Putting Figure 5 and Figure 7 a and b on the same page (or same figure) would be much easier for the readers to see the agreement between the reconstructed maps and satellite observations. Same comment applies to Figure 6 and Figure 7 c and d. 6. Regarding to the assessments of air quality during Youth Olympic, it would be better to show some meteorological parameters during the 3 periods. Technical comments: 1. Page 8, line 27: 'pNO2' should be 'NO2' 2. Page 14, line 12: 'This agree well with the fact' should be 'This agrees well with the fact' 3. There might still be other typos and errors in the manuscript. Please check the entire manuscript carefully.

---

## Author Comment (AC1) · 21 May 2019

We thank reviewer #1 for the comments. Some of these comments are useful for improving our manuscript. We understand that the comments on the scientific content of the manuscript in general are positive, however, several clarifications are necessary. We have addressed the reviewer's comments on a point to point basis as below for consideration. All page and line numbers are refer to the marked-up version of the manuscript.

General comments

[Figure]

In this paper, Chan et al. presented the long-term MAX-DOAS measurements of NO2 and HCHO profiles in Nanjing. The data are used to validate OMI NO2 and HCHO products, discuss the effects of a-priori profiles on OMI retrievals, analyze effects of regional transports, and effects of pollution control measures during the Youth Olympic Games. In general the scientific topic is meaningful, and the MAX-DOAS data quality is well proved. However the authors need to give more deep discussions in many parts to firmly prove the conclusions. Generally I have three major concerns below:

1) Regarding the comparisons with the OMI data in section 3.2, the authors should also show the comparisons of a-priori profile shapes of OMI NO2 retrievals with the NO2 profiles measured by MAX-DOAS for the discussion on the effect of a-priori profiles. In addition, the authors also need to discuss the a-priori effect for HCHO even in the case that a good agreement is found. It is very important to see if good agreements of HCHO profile shapes can been also seen between MAX-DOAS and OMI a-priori.

The underestimation of OMI NO2 VCDs is up to 50% compared to MAX-DOAS data shown in Fig. 2. However the previous study in Wuxi, see Wang et al. 2017, shows the underestimation is âĹij20%. One major difference is that the NO2 product is from NASA in your study, but from DOMINO v2 in Wang et al., 2017. Are there big differences of both OMI NO2 data sets? Why are there big differences? In order to answer the questions, the author needs to do comparisons also with the DOMINO v2 product. Meanwhile DOMINO v2 is an official product which is well known and widely used. In addition the author demonstrates that "Measurements with such large spatial coverage are probably difficult to capture the spatial gradient of NO2 and resulted in an underestimation over pollution hot spots due to the averaging of large OMI footprint. This effect is especially significant over Nanjing, as it is a local pollution hot spot surrounded by rather clean areas". If it is true, the NO2 measured by the MAX-DOAS is dominated by local emission. However the discussion on regional pollution transport in section 3.4, the author concludes that "the air quality of Nanjing is significantly influenced by the air pollution transportation, especially during winter.". The two elaborations are contradictive. Therefore the authors need to carefully discuss the reason of the underestimations of the OMI NO2 data.

Response: The reviewer compares our estimation to previous study in Wuxi, China. We think that these studies cannot be compared directly for several reasons. There are major differences in satellite products used, measurement time and locations. In this study, we compare our MAX-DOAS measurements result to the NASA OMI NO2 standard product version 3 while the previous study in Wuxi used OMI NO2 product produced by KNMI. These products are processed with different algorithms, for example, there are significant differences in the spectral analysis, stratospheric and tropospheric NO2 separation methods, radiative transfer simulation and a priori profiles. Different versions of NASA OMI NO2 products even show a difference up to 40% (Krotkov et al., 2017), let alone the differences between two completely different algorithms. In addition, the differences in measurement time and location also make a big difference in the comparison. Measurements taken at the city center and several tens kilometer away in the suburban can already show a big difference.

To answer the reviewer question, whether the two OMI NO2 products are so different, we have also plotted the DOMINO version 2 data together with the NASA NO2 standard product in Figure 4. The result shows the NO2 VCD from the KNMI product is a factor of 2 higher than that of the NASA product. The result shows that the KNMI OMI product underestimated the NO2 VCDs by $\sim$30% compared to the MAX-DOAS observations. This observation is consistence with the previous study. Further discussion is included in the manuscript (page 13, line 24-34).

We mentioned that regional transportation of pollutants has 'significant' impact on the local air quality which does not implies that regional transport is the 'major' source. Therefore, these sentences do not contradict with each other. Despite the strong local contribution, regional transportation of pollutants can also influence the local air quality. We have further clarified the confusion and rephrase the sentences in section 3.4 (page 18, line 20-21).

2) In section 3.4, the authors used the reconstructed maps to quantitatively validate the satellite maps. Therefore a speculated life time is used to scale MAX-DOAS VCD in the reconstruction of maps. However are the quantitative comparisons reasonable? Because the authors assume that all the pollutants measured by the MAX-DOAS instrument are from emissions in an area corresponding to the starting location of a trajectory. However emissions in different grids along the trajectory route should be mixed up and contribute to the pollutants measured by the MAX-DOAS in reality. The emissions from different distances should be scaled differently. But do we know the proportions of the different emissions? I think the reasonable comparisons of the re-constructed maps with the satellite maps are the relative distributions, but not the absolute values.

The reconstructed maps should depend on the selected backward time of trajectories. The author should show the maps with different trajectory backward time and compare them with the satellite maps in order to see which time is reasonable. And the suitable backward time depends on actual lifetime. Since the lifetime effect is already implied if different backward time is tested, the scaling with lifetime might be not needed and do not give any meaningful results. In addition, transports of pollutants can occur during night time and day time. Lifetime only matters for transports during day time. Nighttime transports can reach a far distance and contribute to concentrations of pollutants during day time. This is another reason why lifetime should not be applied.

The backward propagation method has been applied to long lifetime pollutants and also trace gases measured from MAX-DOAS in previous studies. Some references should be cited in the paper. Meanwhile the sentence "We developed a new technique to assemble the source contribution map using backward trajectory analysis" in the abstract might be inappropriate.

Response: We have to clarify that we did not perform any quantitative comparison between the MAX-DOAS and OMI datasets. We just wanted to remind the reader that there is a large difference in the MAX-DOAS and OMI VCDs. Therefore, the absolute value is expected to be different. Our discussion mainly focused on the spatial distribution of pollutant. We have rephrased the sentences to avoid any confusion (page 18, line 7-9).

I think there might be some misunderstandings with our approach. In our approach, NO2 and HCHO do not decay along the backward trajectory. The assumed lifetimes are only used to compute the weighting factors for the reconstruction of spatial distribution of NO2 and HCHO. As data are more reliable with shorter backward time, therefore, we give higher weight for data with shorter backward time. In order to avoid confusion, we have changed the term 'lifetime weight factor' to 'age weighting factor'. The age weighting approach is useful when multiple trajectories overlapping with each other within a single grid point. In addition, we did not mention that the pollutants measured by the MAX-DOAS are coming from primary emissions. They can also be secondary formed. We have only reconstructed the spatial distribution of pollutants, but not emission maps. In order to avoid the misunderstanding, we have further clarified this issue in the manuscript (page 16, line 6-8, page 17, line 1-2).

We understand that similar backward propagate methods have been used in some other study, we have implemented an age weighting scheme for the spatial distribution inversion, so that it fit better for the application on MAX-DOAS measurements. In order to avoid the confusion, we rephrased the sentences and added references to previous studies which use similar approach (page 15, line 16).

3) In section 3.5, the author compared the pollutants during the Youth Olympic Games with those before and after the event in order to characterize the effect of pollution control measures. Since pollution transports can impact Nanjing as the author demonstrates in section 3.4, the difference of transport conditions in the three periods should also be discussed. Meanwhile the author simply elaborates "As the meteorological conditions are very similar during the three periods". I think the author has to show wind fields, trajectories, precipitations, and temperatures in the three periods in order to convince the readers. Near-surface concentrations of the pollutants should be also derived from the MAX-DOAS profile inversion. Since near-surface concentrations

should be mainly dominated by the local emission, but VCDs (AODs) contain contributions of pollutant transports. Therefore it is also meaningful to include the comparisons of the surface concentrations as VCDs shown in Fig.9.

Response: We have added the meteorological measurements such as temperature, wind speed and wind direction to support the discussion. The meteorological data are shown in Figure 10. A brief description of the meteorological data is added to section 2.3.

In addition, surface mixing ratios of NO2 and HCHO are now supplemented in Figure 9c. A more detailed discussion regarding the reduction of surface NO2 and HCHO concentration is included in section 3.5 (page 20, line 9, page 21, line 1-5).

Specific Comments:

1) P4 L3: A reference should be given for QDOAS. Please clarify which of the two wavelength ranges is used for NO2 and HCHO?

Response: We have supplemented the fitting range in the text (page 5, line 1-2) and cited reference for the use of QDOAS software (page 4, line 2).

2) P4 L4: Please clarify the reference spectrum is the zenith measurement in individual elevation scan or around noon time?

Response: We use the zenith spectrum taken in the same measurement cycle as reference in the analysis. This information is now supplied in the manuscript (page 5, line 2).

3) Table 1: Do you determine the DOAS fit settings based on sensitivity studies (which are not shown) or previous studies? If you determined them based on previous studies, some references should be given. In addition, do you apply the wavelength dependent Ring suggested by Wagner et al., 2009? If not, please discuss why the additional Ring is not needed in your analysis. Wagner, T., Beirle, S., and Deutschmann, T.: Three-dimensional simulation of the Ring effect in observations of scattered sun light using

Monte Carlo radiative transfer models, Atmos. Meas. Tech., 2, 113-124, 2009.

Response: The DOAS fit settings are taken from QA4ECV project and have been adopted for the CINDI-2 campaign. We have referred these DOAS fit settings to the previous study (page 5, line 7-10).

4) Section 2.1.2: examples of DOAS fits should be shown, especially for HCHO, in order to convince the quality of HCHO analysis.

Response: We have added an example of the DOAS fit as Figure 1. A brief description is also supplemented in the manuscript (page 5, line 10-12).

5) P4 L17-19: How do you filter the data under continuous clouds when the variability of O4 dSCDs are not large?

Response: Our cloud filtering approach is based on the analysis of the time series of O4 DSCD measured at each elevation. As O4 DSCDs are expected varying smoothly with time under clear sky condition, rapid change of O4 DSCDs are likely related to the present of cloud in the atmosphere. Therefore, we applied a high pass filter to the O4 DSCD time series to screen out cloud contaminated observations. The only limitation of this cloud screening algorithm is that the algorithm cannot distinguish continuous and homogeneous cloud condition. However, it is rare that the cloud does not change for a long time (within an hour) and the cloud layer is homogeneous for all viewing directions. We have also tried the color index method for cloud screening. However, the color index method tends to filter data will high aerosol load as the sky is whiter under high aerosol load condition. In addition, the aerosol load are usually high in Nanjing, the color index method identifies most of these high aerosol data as cloud contaminated. Therefore, we use the former high pass filter method for cloud screening in this study. We have supplemented the limitation of the cloud screening algorithm in section 2.1.1 (page 6, line 2-5).

6) P5 L15: Since O4 VCD can systematically vary during a year due to variations

of temperature and pressure, as Wagner et al. (2018 AMT) demonstrated, the phenomenon can explain the scaling factor in many places. How do you consider the variation of temperatures in the retrievals of aerosols? If you don't consider it, a discussion on the uncertainties due to the effect has to be given.

Response: We use the U.S. standard mid-latitude atmosphere profiles for winter (January) and summer (July) and temporally interpolated to each month of the year for the radiative transfer simulation. This information is now supplemented in the manuscript (page 6, line 30-31)

7) P6 L11: How do you determine the single scattering albedo and asymmetry parameters, and also Ångström coefficient? The parameters can significantly change in the long-term measurements, uncertainty estimations of aerosol results due to the parameters should be given in the paper.

Response: These values are inherited from previous study. In order to investigate the uncertainty caused by the fixed set of aerosol optical properties, we performed sensitivity analysis using aerosol optical properties from the sun-photometer measurements in Nanjing and AERONET station ~150km southeast of the measurement site. The result shows that the uncertainty caused by Ångström coefficient, single scattering albedo and asymmetric parameter is ~2%, 1.5% and 4% respectively. This information is supplemented in the manuscript (page 7, line 27-30, page, line 29-31, page 11, line 1-5).

8) P6 L16-18: How do you determine the wavelengths of the AMF simulations of O4, NO2, and HCHO?

Response: We choose 360nm for the simulation of O4 DSCDs simply due the strong absorption at this wavelength. This wavelength is also commonly used in many studies for O4 simulations. As the NO2 DSCDs are also retrieved in the same fitting window, therefore, we adapted the same wavelength of 360nm for NO2 AMF simulation. For HCHO retrieval, our choice of AMF simulation wavelength of 340nm is close to the

center wavelength of the DOAS fitting window of 342nm. This wavelength is also commonly used in HCHO retrieval, e.g., De Smedt et al., 2018.

9) P6 L20: How do you deal with the NO2 above 3km? The considerable amount of NO2 at high altitudes might also impact retrievals of NO2 below 3km.

Response: The MAX-DOAS measurements are not sensitive to higher altitudes. Therefore, we assume that the NO2 profile follows the US standard atmosphere. We have supplied this information in the manuscript (page 7, line 32).

10) Section 2.1.3: Figures of comparisons of measured dSCDs and modeled dSCDs for profile retrievals should be shown in the manuscript or supplement to show the convergence of the profile retrievals.

Response: We have added an example of aerosol, NO2 and HCHO profile retrieval in Figure 2.

11) Section 2.3: The overpass time of OMI should be given.

Response: We have now added the overpass time of OMI (page 9, line 22-23).

12) P8 L13-14: Can the constraint of a-priori profile contribute to the underestimations? In order to show this, comparisons of measured dSCDs and modeled dSCDs are needed.

Response: We have added an example of aerosol, NO2 and HCHO profile retrieval in Figure 2 which included measured and modeled DSCDs.

13) P8 L28: As I know, there are not domestic heating systems in Nanjing since it is in the south of Huai River.

Response: The reviewer is partly correct. There is no centralized heating system in Nanjing, but some of the new buildings are equipped with individual heating system with typically run on natural gas or electricity. Although the domestic heating emissions from southern part of China are smaller than that of the northern China, their contribution can still be significant. In order to avoid any confusion, we have rephrased the sentence in the manuscript (page 13, line 5-6).

14) P10, L1: The underestimation of OMI NO2 compared to MAX-DOAS is not consistent with Wang et al., 2017. The underestimation here is much stronger.

Response: See response to general comment 1.

15) P13, L6: Oxidation rate of VOCs to HCHO is also stronger in summer than in winter. The variations of oxidation rate can also contribute to seasonal pattern of HCHO. And secondary sources of HCHO are significant. The seasonal pattern of HCHO might be due to contributions of biogenic emissions of precursor VOCs. The sentence should be modified.

Response: We have revised the expression of the sentence and included the cause of higher oxidation rate of VOCs in summer (page 14, line 14-15)

16) P13, Figure 5: The color scale of subfigure (a) should be changed to allow seeing the gradient more clearly. As I elaborated in General comment (2), the relative distribution is much more important than the absolute values.

Response: We have adjusted the color scale of Figure 6 and 7.

17) P13, L16: Since you calculate the trajectories in each altitude grids of MAX-DOAS profiles, how do you combine the trajectories with the profiles? Do you assign partial columns in each vertical grid to different grid points in the map along trajectories at individual altitudes? The procedure need to be clarified.

Response: We have supplemented a more detailed description of the spatial reconstruction procedure (page 16, line 2-4).

18) P14, L5-6 how do you determine the lifetime and backward time? The question is corresponding to the general comment (2).

Response: See response to general comment 2.

19) P14, L12: As I demonstrate in comment (2), the quantitative comparisons with OMI data are not reasonable.

Response: See response to general comment 1.

20) P14, L19: Since HCHO is dominated by the secondary formations from VOCs, which have a long lifetime, therefore VOCs might be transported to a far distance and contribute to local HCHO concentrations. Therefore transport effects on HCHO might be even larger if the transports are from far distance. The backward time of trajectories of 6 hour might be not long enough in the reconstruction of HCHO maps. Following my general comment (2), I suggest you to generate the maps with different backward time of trajectories and compare the relative distributions with OMI maps.

Response: As discussed in response to general comment 2, we are not trying to re-construct source map but the spatial distribution of HCHO. In addition, the lifetime is just used for the calculation of weighting factor. This approach is useful when multiple trajectories overlapping with each other within a single grid point. Of course we have looked into map with different backward time. As expected maps created with shorter backward time correlates better with OMI observations, but then the spatial coverage are very limited. In order to get a balance between having better spatial coverage and the reliability of the reconstructed pollution maps, these numbers are used in this study. This information is now added in the manuscript (page 17, line 7-8, page 18, line 1).

---

## Author Comment (AC2) · 21 May 2019

We thank reviewer #2 for the useful comments. These comments are helpful for improving our manuscript. We understand that the comments on the scientific content of the manuscript in general are positive, however, several clarifications are necessary. We have addressed the reviewer's comments on a point to point basis as below for consideration. All page and line numbers are refer to the marked-up version of the manuscript.

The manuscript entitled 'MAX-DOAS measurements of tropospheric NO2 and HCHO in Nanjing and the comparison to OMI observations' by Chan et al. presented a long term

[Figure]

MAX-DOAS observations of atmospheric nitrogen dioxide (NO2) and formaldehyde (HCHO) in Nanjing. The MAX-DOAS measurements were validated by comparing to sun-photometer observations. The authors then used the MAX-DOAS data for the validation of the NASA's OMI NO2 and HCHO products. OMI observations in general show a good agreement with the ground based observations. A discussion of a priori profile on the satellite retrieval is also presented. The MAX-DOAS data is also used for the investigation of regional transportation of pollutants and for the assessment of air quality during the Youth Olympic Game in Nanjing. The study is in general well written and scientifically interesting for the community. Therefore, I recommend publishing the manuscript after addressed the following minor comments.

Minor comments:

1. Although the agreement between OMI and MAX-DOAS HCHO observations is already very good, it is still interesting to see the effect of MAX-DOAS profiles being used for OMI HCHO VCDs retrieval. I understand that there is a large fraction of HCHO above the MAX-DOAS retrieval height compared to NO2, using the MAX-DOAS profile would result in a larger OMI HCHO columns. This is also relevant for other MAX-DOAS satellite comparison studies.

Response: Following the reviewer's comment, we have added the OMI HCHO VCDs retrieved with MAX-DOAS measurements as a priori for comparison. The results are indicated in Figure 5. A more detailed discussion is added to the manuscript (page 14, line 19-26).

2. The authors mentioned on page 11 that it is difficult to quantify the effect of spatial inhomogeneity of NO2 on the satellite data comparison due to lack of high spatial resolution data. I am wonder if the MAX-DOAS is still measuring? If yes, then it would be useful to compare the latest measurement to TROPOMI observations. As TROPOMI provides much higher spatial resolution data, these datasets are very useful for the quantification spatial gradient effect on satellite to ground based measurement

comparison.

Response: Unfortunately, the MAX-DOAS measurements only cover a period from April 2013 to March 2017 while TROPOMI was launched in October 2017. The MAX-DOAS is under maintenance after March 2017 till now. Therefore, there is no overlap between the MAX-DOAS and TROPOMI data. We will try to the compare the MAX-DOAS data to TROPOMI observations when the MAX-DOAS is back online.

3. The reconstruction of spatial distribution NO2 and HCHO from MAX-DOAS measurements using trajectories simulations are particularly interesting. However, the description of the method is a bit too brief. The author should include a more detail description.

Response: We have supplemented a more detailed description of the spatial reconstruction procedure (page 16, line 2-8).

4. In addition, do the authors try other lifetime weight factors in this study? Are the weighting factors determined by fitting to satellite data or the authors just select a realistic one? This can be relevant for other similar studies.

Response: The assumed lifetime is only used for the calculation of the weighting factor and the weighting factor is only useful when multiple trajectories overlapping with each other within a single grid point. We have tried different lifetime and backward time to reconstruct the spatial distribution map of NO2 and HCHO. As expected maps created with shorter backward time correlates better with OMI observations, but then the spatial coverage are very limited. In order to get a balance between having better spatial coverage and the reliability of the reconstructed pollution maps, these number are used in this study. This information is now added in the manuscript (page 17, line 7-8, page 18, line 1).

5. Putting Figure 5 and Figure 7 a and b on the same page (or same figure) would be much easier for the readers to see the agreement between the reconstructed maps

and satellite observations. Same comment applies to Figure 6 and Figure 7 c and d.

Response: We followed the reviewer's comment and combined Figure 5 and Figure 7 a and b as a single figure. Same procedure is also applied to Figure 6 and Figure 7 c and d.

6. Regarding to the assessments of air quality during Youth Olympic, it would be better to show some meteorological parameters during the 3 periods.

Response: We have added the meteorological measurements such as temperature, wind speed and wind direction to support the discussion. The meteorological data are shown in Figure 10. Description of the meteorological data can be found in section 2.3.

Technical comments:

1. Page 8, line 27: 'pNO2' should be 'NO2'

Response: We have corrected the typo.

2. Page14, line 12: 'This agree well with the fact' should be 'This agrees well with the fact'

Response: We have corrected the grammatical mistake.

3. There might still be other typos and errors in the manuscript. Please check the entire manuscript carefully.

Response: We have proofread the manuscript carefully to avoid any typo and error.

---

## Author Response (AR2)

Response to Co-Editor

We thank the editor for carefully go through the manuscript and provided useful comments. We have addressed the editor's comments on a point to point basis as below for consideration. All page and line numbers refer to the marked-up version of the manuscript.

1. As requested by the reviewer, better to compare the MAX-DOAS NO2 profile shape with a-priori used for OMI satellite retrievals, or to analyze the effect of using MAX-DOAS profile shape on OMI retrievals. Although the authors commented that they did not perform any quantitative comparison between the MAX-DOAS and OMI VCDs, I believe that the value of this manuscript is present in this particular point.

Response: Seasonal average of a priori profiles of NO2 and HCHO are shown in Fig. 6. Discussion of the influence of a priori profile on the VCD retrieval is also included in the manuscript (page 14, line 6-8, page 15, line 17-20).

page 7, line 33: What are the NO2 and HCHO from the U.S. standard atmosphere? Give a reference.

Response: A reference has been added (page 9, line 3).

minor points:

page 2, line 5 (change-track version): The emission of NOx

Response: Corrected (page 2 line 5).

page 2, line 12: Contribution of HCHO for secondary aerosol formation is believed to be minor.

Response: We have revised the statement and mentioned that the contribution from HCHO is relatively small (page 2, line 13-14).

page 14, line 19. quantify

Response: Corrected (page 15, line 11).

page 16, line 7. I do not understand what is meant by "Data are more reliable with shorter atmospheric lifetime."

Response: We have revised the sentence to 'In addition, model error accumulates over time. Results with shorter simulation time (or backward time) are considered more reliable.' (page 16, line 16-17, page 17, line 1).

page 18, lines 6ff. The OMI data are from NASA retrievals. Better to specify as the authors now use KNMI retrievals too. Rephrase "As the MAX-DOAS NO2 VCDs are on average 3 times higher than the OMI data" from line 7, as almost similar sentence is present just before.

Response: We have specified the source of OMI data (page 20, line 3). The repetitive sentence has also been rephrased (page 20, line 4-7).

page 22, line 6. Another source ... INCLUDES difference in spatial ... would be better, as yet another possibility like aerosol perturbation might also be present.

Response: We followed the editor comment and rephrased the wording (page 24, line 26).

Response to reviewer #1

We thank reviewer #1 for the useful comments. We understand that the comments on the scientific content of the manuscript are in general positive, however, several clarifications are necessary. We have addressed the reviewer's comments on a point to point basis as below for consideration. All page and line numbers refer to the marked-up version of the manuscript.

Most of my comments have been well answered. And more details and deep discussions have been added in the revised manuscript. However there are still some points which are not well considered:

1) Although I asked the author to show the comparisons of a-priori profiles and profiles measured by the MAX-DOAS in the major comment 1 of the previous review, a proper reply the was not given by the author. Please reply for this point.

Response: Seasonal average of a priori profiles of NO2 and HCHO are shown in Fig. 6. Discussion of the influence of a priori profile on the VCD retrieval is also included in the manuscript (page 14, line 6-8, page 15, line 17-20).

2) In the revised draft, the author applied the MAX-DOAS HCHO profile in the calculations of OMI AMFs. And the author attributed the overestimations of the corrected OMI HCHO VCDs compared to MAX-DOAS VCDs to "the MAX-DOAS measurements are not sensitive to HCHO at higher altitude". This is a strong claim on the credibility of MAX-DOAS HCHO measurements. OMI tropospheric HCHO SCD retrievals might also play a role here since some post corrections might be applied. If the author will insist on the claim, a study on typical proportions of HCHO located at upper troposphere in total HCHO columns and a sensitivity study of MAX-DOAS HCHO profile retrievals need to be done. I suggest the author modify their statement.

Response: We followed the reviewer suggestion and rephrased the sentences to 'Higher HCHO VCDs retrieved from OMI using MAX-DOAS profiles as a priori is mainly due to ignoring HCHO in the upper altitudes. The a priori show a considerable amount of HCHO above 3km especially during summer, while the MAX-DOAS only reports HCHO mixing ratios up to 3km above ground level.' (page 15, line 17-20).

3) Regarding my second major comment, the author might not understand my argument. My argument is whether or not the lifetime weighting factor is meaningful to improve the backward propagation method. Both the trajectory backward time and lifetime can impact the

weighting factor based on your equation 1, but both are arbitrarily determined without a reasonable sensitivity study shown in the draft. Meanwhile if the NO2 life time of 6 hours is assumed, do the trajectories of less than 12 hours also play a role, e.g. 1, 2, 3… hours? And how is the mixing of air parcels with different backward time considered in your approach?

For HCHO, the author assumed a shorter backward time, however as we see in Fig. 8c, the HCHO map observed by OMI is much smoother than OMI NO2 map shown in Fig. 7. Although HCHO lifetime is shorter than NO2, secondary formation of HCHO from its VOC precursors along regional transports can contribute to your measurement site. In another word, HCHO which the MAX-DOAS instrument measured was not all from primary emissions, but also from secondary formations. This effect means a larger backward time than 1 hour should be applied. The author might reference the sensitivity test on the effect of trajectory backward time on reconstructed maps shown in the recent paper of Wang et al., 2019 (which is cited in your draft). They applied the similar approach and determined the backward time reasonably by comparing reconstructed maps with satellite maps.

Response: I fully understand the reviewer concern and our answer is in the following. As we have mentioned in the previous reply that the assumed lifetime is only used to calculate the age weighting factor. The age weighting factor is to take into account that pollutants at the same spot are being transported to the measurement site at different time with different wind speed (also backward time). As model error accumulates over time, results with shorter backward simulation time are expected to be more reliable. In addition, the measured pollutant concentrations are less related to the pollutant concentrations far from the site which take longer time to reach the measurement site. It is also the reason that the calculation limited to certain simulation (backward time). A further explanation is supplemented in the discussion (page 16, line 16-17, page 17, line 1). The assumed lifetime or the age weighting factor play no role in the calculation if there is no overlapping trajectory. Of course, we have tried different lifetime (or age weighting factor) in our calculations. However, we think the sensitivity analysis results are less important. Therefore, we did not include it in the manuscript in the first place. The sensitivity analysis result is now included in the manuscript (Fig. 10).

Fig. 10 shows the restructured NO2 map with different assumed lifetime which is labeled on each figure. The spatial correlation between the reconstructed map and OMI observations are also indicated. It can be seen in the sensitivity analysis that the accuracy of the lifetime does not show a strong impact on the reconstructed spatial pattern, as the assumed lifetime is only used for the calculation of the age weighting factor. The assumed life time is used in this study based on the consideration of get a balance between having better spatial coverage and the reliability of the reconstructed pollution maps. The discussion of the sensitivity analysis result is also included in the manuscript (page 17, line 7-8, page 18, line 1-7).

Regarding to the reconstruction of HCHO distribution, as we have mentioned before, we are not reconstructing an emission (or source) map but just the spatial distribution of HCHO. Smoother spatial distribution of HCHO is likely related to its source characteristic. HCHO is mainly secondary formed with major sources of HCHO (and its precursors) from biogenic

emissions. The emission sources of these precursors are mostly area source, i.e., vegetation, while NO2 is mostly emitted from point source, i.e., power plant and traffic emission. Therefore, the spatial distribution of HCHO is expected to be smoother compared to that of NO2.

We agree with the reviewer that the HCHO measured by the MAX-DOAS are partly attributed to the secondary formation from its precursors. However, the objective of the backward propagation study is to reconstruct the spatial distribution of HCHO rather than its precursors. Therefore, a short lifetime as well as backward time should be used. Otherwise, the reconstructed maps are more related to HCHO precursors rather HCHO itself. A further clarification is added to the manuscript (page 20, line 12-15).

4) Regarding effects of meteorology on pollutants around the Youth Olympic Games, the author wrote "as the meteorological conditions are very similar during the three periods", but it is not true for temperature. Figure 10b indicates that the temperature in the pre-Olympic period is higher than those in the Olympic and post-Olympic periods. Temperature can impact secondary formation of HCHO. The effect might contribute to the higher HCHO concentrations in the pre-Olympic period than those in the other two periods. The author needs to consider the effect.

In addition, as I suggested in the major comment 3 of the previous review, "difference of transport conditions in the three periods should also be discussed". The statistics of trajectories might be needed to show for the discussion. However a reply to this point was not given for this point.

Response: We agree with the reviewer that higher temperature would bring more HCHO. The meteorological data indicates that the ambient temperature is decreasing with time as season changing from summer to autumn. The HCHO concentrations measured during the Youth Olympic are expected to be higher than that measured in the post-Olympic period if the anthropogenic emissions are unchanged. However, the measurements shows the HCHO concentrations are slightly lower during the Youth Olympic compared to the post-Olympic period. The results imply that the emissions of HCHO (and its precursors) are reduced during the Youth Olympic Games. The explanation is included in the manuscript (page 23, line 8-9, page 24, line 1-5).

The differences in transport conditions during the three periods are partly reflected by the wind speed and wind direction data indicated in Fig. 11. Analysis of trajectories does not show significant difference among the three periods. In response to the reviewer question, we have added the trajectory plot for all 3 periods (Fig. 13). In addition, a brief discussion of the trajectory analysis is supplemented in the manuscript (page 22, line 2-4).

5) Please add the information of the reply to my specific comment 8 into the draft.

Response: We have included the reply in the manuscript (page 7, line 6-7, 27-33).

[revised manuscript text omitted]

---

## Author Response (AR3)

Response to reviewer #1

We thank reviewer #1 for the useful comments. We have addressed the reviewer's comments on a point to point basis as below for consideration. All page and line numbers refer to the marked-up version of the manuscript.

The authors well considered the comments and suggestions. One remaining issue might need to be clarified. For the sentence (L4-5, P18) "Backward trajectories are calculated over 6 h for summer measurements and 12 h for winter measurements", it is not clear if the authors used trajectories of shorter than 6h in summer and 12h in winter. If yes, since the age weighting factor depends on the backward time, therefore the backward time determines the dominant trajectory through the summation of values according to individual trajectoris with different backward time overlapped in a map grid.

Response: The reviewer is correct. We assign the measurement values to the grid points along the backward trajectories with weighting depending on its simulation time. As the simulation time (or backward time) is changing along the trajectory, the weighting of each point along the trajectory is of course different. We have further clarified this issue in the manuscript page 18, line 7-9.